# Percolation transition prescribes protein size-specific barrier to passive transport through the nuclear pore complex

David Winogradoff [1,2,4], Han-Yi Chou[1,4], Christopher Maffeo [1,3] & Aleksei Aksimentiev [1,2,3] ✉

Nuclear pore complexes (NPCs) control biomolecular transport in and out of the nucleus. Disordered nucleoporins in the complex's pore form a permeation barrier, preventing unassisted transport of large biomolecules. Here, we combine coarse-grained simulations of experimentally derived NPC structures with a theoretical model to determine the microscopic mechanism of passive transport. Brute-force simulations of protein transport reveal telegraph-like behavior, where prolonged diffusion on one side of the NPC is interrupted by rapid crossings to the other. We rationalize this behavior using a theoretical model that reproduces the energetics and kinetics of permeation solely from statistics of transient voids within the disordered mesh. As the protein size increases, the mesh transforms from a soft to a hard barrier, enabling orders-of-magnitude reduction in permeation rate for proteins beyond the percolation size threshold. Our model enables exploration of alternative NPC architectures and sets the stage for uncovering molecular mechanisms of facilitated nuclear transport.

The nucleus secludes the genetic material of a eukaryotic cell to ensure the fidelity of transcription, replication, and gene regulation processes[1–5]. Nuclear pore complexes (NPCs) perforate the nuclear envelope, a double lipid bilayer that encases the nucleus, providing a passage for biomolecular traffic in and out of the nucleus. Water, ions and small biomolecules, up to ~5 nm in diameter, can pass through an NPC largely unimpeded. Unassisted transport of larger molecules is blocked[6–8], but, when combined with nuclear transport factors, cargoes up to 40 nm in diameter can traverse an NPC[9,10], fueled by a RanGTP/GDP cycling system[11,12]. The selectivity of nuclear pore transport is attributed to the properties of nucleoporin proteins (nups) that form a barrier to diffusion through the NPC's central channel[13,14]. Many of such nups have repeating phenylalanine-glycine (FG) motifs[15] that form hydrophobic, highly dynamic[16], and intrinsically disordered domains[17] known to interact with nuclear transport factors[18,19]. Medically, disturbances in nuclear transport can lead to a number of human

diseases, including cancer, viral infections, and neurodegenerative conditions[20]. Nuclear transport is also a key target for emerging gene therapy[21].

Essential to the regulatory role of the NPC, its central mesh serves as a filter cutting off passive transport through the NPC above some molecular weight or geometrical size threshold. Theoretical studies have investigated passive diffusion through model narrow channels, showing that the presence of an attractive site within the channel can greatly increase the diffusive flux[22] and enable transport selectivity[23]. For a given concentration of solutes, the attractive interactions can be tuned to optimize the transport[24], although, at high concentrations, the highest flux is achieved when an attractive potential is replaced by a barrier[25]. The transport becomes even more nuanced when multiple binding sites are present within the channel[26]. Relating to nuclear transport specifically, early experiments[27,28] observed proteins as large as 10 nm in diameter to passively diffuse across an NPC. More recent

[1]Department of Physics, University of Illinois at Urbana-Champaign, Urbana, IL 61801, USA. [2]Center for the Physics of Living Cells, University of Illinois at Urbana-Champaign, Urbana, IL 61801, USA. [3]Beckman Institute for Advanced Science and Technology, University of Illinois at Urbana-Champaign, Urbana, IL 61801, USA. [4]These authors contributed equally: David Winogradoff, Han-Yi Chou. ✉e-mail: aksiment@illinois.edu

experiments[6,8,29,30] established the absence of a sharp size threshold for the passive diffusion, although 5 nm diameter[29] or a 40–60 kDa mass are often cited as effective size and molecular weight thresholds, respectively. Fluorescent tracking experiments revealed that, given enough time, proteins as large as 200 kDa and 8 nm in diameter can cross an NPC unassisted by transport factor[6], implying that passive transport of large cargoes through an NPC is not impossible, but is rather impractical, from a cell biology point of view.

Computational studies play an important role in evaluating possible mechanisms of nuclear pore transport by providing insight into microscopic processes not readily accessible to experiment. Molecular transport through the NPC has been examined within the framework of the kinetic theory[31], showing how selectivity can arise from the competition for the limited space inside the channel. Polymer theory methods have been applied to evaluate the effect of the cargo size on the configurational entropy of a polymer brush grafted to a model channel[32], suggesting that the size-dependent selectivity may originate from an entropic effect. Free-energy models have been developed to show how molecularly divergent NPCs in different biological species can perform essentially the same function[33] and how a complex free-energy profile may arise from electrostatic and hydrophobic residues within the NPC mesh[34], including local electrostatic polarization of the nup domains[35]. Coarse-grained (CG) molecular dynamic simulations have been combined with a theoretical model to determine the effect of cohesive interactions on protein transport[36], characterize passive transport through a model NPC system[8], and to directly evaluate the free-energy barrier for translocation through a synthetic channel decorated with disordered nup proteins[37,38], finding the local interaction with the mesh to enhance transport of larger particles[39,40].

Here, we combine a computational model of an experimentally derived NPC structure with a percolation transition analysis to determine the physical origin of the barrier to passive diffusion of globular proteins across an NPC. Through brute-force simulations of passive diffusion, we directly characterize the effect of protein size on the passive diffusion rate. We rationalize the observations by examining an ensemble of configurations realized by the nup mesh in the absence of any proteins, arriving at a general method for estimating the free-energy barrier and the transport rates. Using our theoretical framework, we discover a crossover in the scaling behavior of the passive diffusion rate with respect to the protein size and identify a percolation transition to be at the origin of the crossover. We confirm our findings by carrying out computational analysis on another experimentally derived NPC structure. Our work sets the stage for computational characterization of passive transport through NPC variants differing in their composition, stoichiometry, and physical dimensions, providing a means to connect variations in NPC structure[41,42] and plasticity[43,44] to the NPC's function as the gatekeeper of nuclear transport.

## Results

### A computational model of a composite NPC structure

Starting from the experimentally derived composite structure of an NPC[45] (referred hereafter as the Lin2016 structure), we developed a computational model that included a custom grid-based potential representation of the nuclear envelope and of the protein scaffold and a one-amino-acid-per-bead representation of the disordered FG-nup mesh (Fig. 1a, b). The shape of the nuclear envelope was derived from a cryo-electron tomography reconstruction (EMD-3103)[46], whereas the scaffold potential was derived from the composite structure[45]. Matching the stoichiometry of a composite NPC structure[45], our computational model included 32 copies each of Nsp1, Nup49, Nup57, Nic96, and Nup145N proteins, with their disordered mesh parts generated as a self-avoiding random walk starting from the point anchoring each nup to the scaffold. We chose not to include cytoplasmic filaments or the nuclear basket in our model, as their

structures remain poorly characterized by experiment. The model was simulated using an in-house developed software ARBD[47]. The interactions between the beads representing the disordered mesh were described using a model developed by the Onck laboratory[48,49]. Custom potentials described interactions of the FG-nup beads with the NPC scaffold and the nuclear envelope, see Methods for details.

Two 7.5 millisecond replica simulations of our computational model characterized the highly heterogenous and dynamic ensemble of conformations adopted by the network of disordered FG-nup proteins. Figure 1a and Supplementary Movie 1 illustrate one simulation trajectory. The simulations revealed the formation of transient channels within the mesh network, connecting the two compartments on the opposite sides of the NPC complex. One such channel is clearly visible in the 5000 µs overhead snapshot in Fig. 1a. The local density of the FG-nup mesh, averaged over the two simulation trajectories and over the symmetry of the NPC complex (Fig. 1b), displays a donut-shaped region of high (>55 mg/mL) local density surrounding the scaffold's inner ring, and reduced density within the central channel. In comparison to earlier computational studies of model NPC systems[49,50] (Supplementary Fig. 1a), our local FG-nup density map has similar values in the middle of the central channel but shows much lower values near the anchor points, which we attribute to differences in both the scaffold dimensions and the nups' stoichiometry. Our density map is comparable to that reported in a more recent computational study[35] (Supplementary Fig. 1a). We find the spatial distribution of the hydrophobic residues of the FG-nups (Supplementary Fig. 2d) to follow the overall FG-nup density, similar to observations reported in ref. 49.

Averaging the local densities of individual nups (Fig. 1c), we find Nsp1, Nup49, Nic96, and Nup57 to extend into the central channel, with Nsp1 extending the farthest. At the same time, Nup145N is seen to primarily fill the space between the inner and outer rings of the NPC, whereas Nic96 partially fills the cavities present in the structured protein scaffolding, consistent with its known role in holding the NPC assembly together[51]. Within the context of the NPC, FG-nups condense together at the nanoscale, which we consider to be related to, but distinct from, the liquid–liquid phase separation that drives the formation of membraneless organelles[52,53]. Constrained by the NPC scaffold with interactions programmed by their specific sequences, FG-nup domains are highly connected together and undergo what Huang et al.[35] refer to as "nanophase separation".

A property of the FG-nup mesh that is often discussed with regard to the mechanism of nuclear transport is the extent to which nups connect together to generate a mesh. Analysis of our simulation trajectories shows that about 65% of all FG-nups chains are connected, on average, forming at least one contact with a residue from another nup. A representative configuration of the FG-nup mesh (Fig. 1d), shows that the nups tethered to the NPC's inner ring are almost all connected: over 80% of Nup57, Nup49, and Nsp1 form at least one contact with another nup (Fig. 1e). Nup145N is the least likely to form interchain contacts with other nups because of its anchor position that is far from the midplane dividing the NPC into two halves along its pore axis, whereas Nic96 exhibits the greatest variability of its connectedness. In all, a single nup can form up to twenty interchain contacts (Supplementary Fig. 2a). The differences among nup species is consistent with Nup145N and Nic96's classification as "adaptor nups," distinct from Nsp1, Nup49, and Nup57, which are considered to be the "channel nups." Hydrophobicity is hypothesized to play an important role in forming the FG-nup mesh. The percentage of all inter-nup contacts that contain a hydrophobic residue, or indeed any other specific residue type, appears to largely be prescribed by the abundance of that particular type (Supplementary Fig. 2b). We note that the FG-nup segments present in our Lin2016 model all have a relatively low charge-to-hydrophobicity ratio, i.e., below 0.3 (Supplementary Fig. 2c). Our results suggest that FG-nups within the NPC central channel are mostly connected with each other through transient interactions.

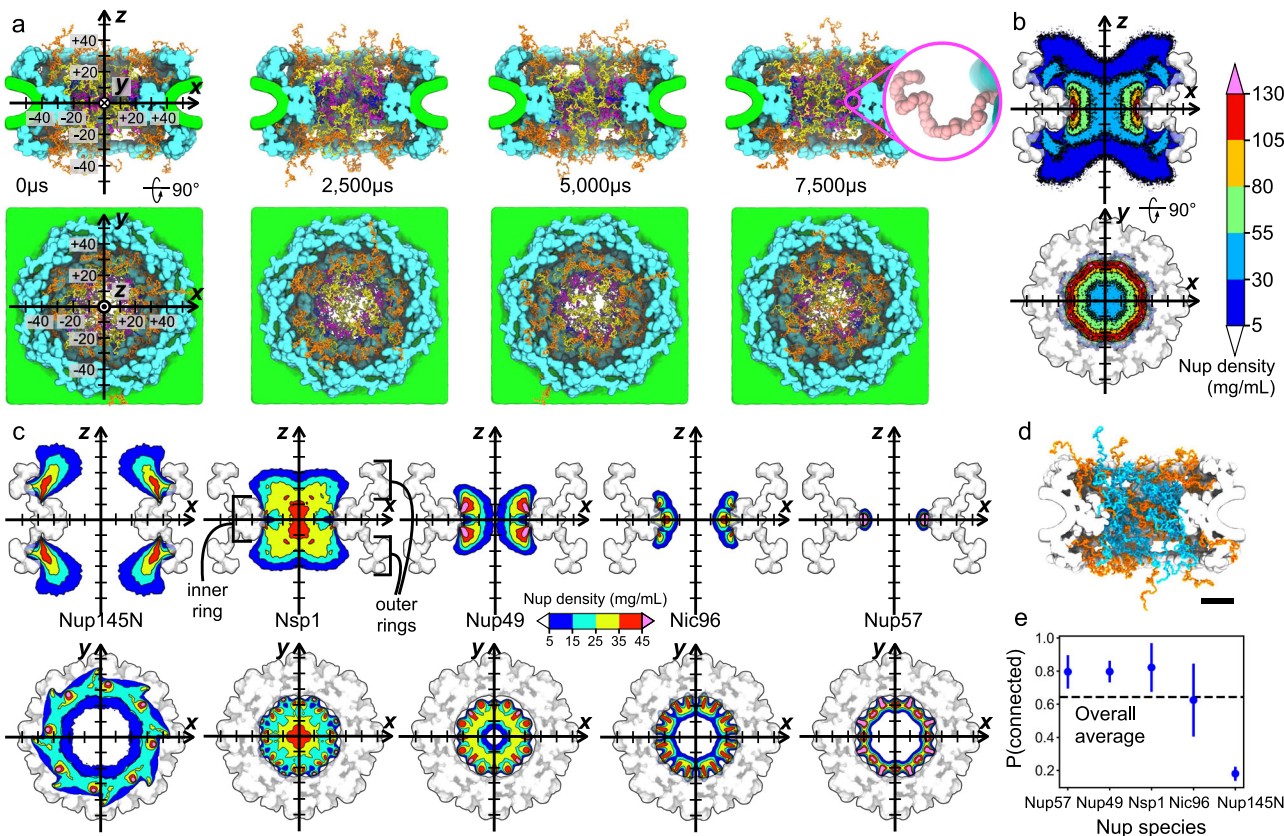

**Fig. 1 | Computational model of a composite NPC structure. a** Equilibration simulation. The computational model consists of a nuclear envelope (green), structured protein scaffold (cyan), and disordered FG-nups mesh consisting of 32 copies each of Nup145N (orange), Nsp1 (yellow), Nup49 (magenta), Nic96 (dark blue), and Nup57 (pink). Labels for the coordinate system (on the left) are in nm. Inset shows a magnified view of Nup57; each bead represents one amino acid. **b** Cross sections of the average FG-nup amino-acid density. The 3D density map was generated by averaging instantaneous configurations of the computational model every 0.1 μs over the final 6 ms fragments of two independent 7.5 ms simulations. The side and top view cross sections were additionally averaged along the $y$ and $z$ coordinate, respectively, within the [−7.5, +7.5] nm range. **c** Average density maps of individual FG-nup species generated using the same protocols as the density map shown in panel **b**. Black brackets define the "inner ring" and "outer rings" regions of the NPC scaffold. **d** Representative configuration of the NPC mesh, where individual FG-nups making at least one contact with another FG-nup are shown in blue and those without such contacts are shown in orange. Scale bar, 20 nm. **e** The fraction of nups forming at least one interchain contact, by species. Data presented as mean ± SD, based on $N = 65{,}000$ frames. The dashed line shows the fraction averaged over all species. An interchain contact was defined as having two residues from different chains within 0.8 nm of one another.

## Passive diffusion of individual proteins across the NPC

To examine the process of passive diffusion across the NPC, we constructed 26 simulation systems, each containing, in addition to our computational model of the NPC, a single protein represented as a rigid body that interacted with the NPC via the same potentials as beads of the FG-nup mesh, see Methods for details. Thirteen unique protein species were modeled, ranging in molecular mass from 5 to 145 kDa (Fig. 2a and Supplementary Table 1). To increase the probability of a protein's encounter with the FG-nup mesh, the center of mass (CoM) of the protein was subject to a confinement potential (Fig. 2b), which reduced the volume available for protein diffusion to a cylinder co-axial with the pore of the NPC. Our rigid-body approximation of the protein preserved the protein shape at the one-bead-per-residue resolution (Fig. 2c). The translational and rotational diffusion constants of each protein were separately defined along the principal component axes of the corresponding rigid body and used to simulate translational and rotational displacements via a Brownian dynamics algorithm. Two independent 6000 μs simulations were performed for each protein species, differing only by the initial location of the diffusing protein (Fig. 2b). The protein diffusion simulations were also repeated using a narrower, 25 nm-radius confinement potential (Supplementary Fig. 3).

In a typical simulation, a protein was observed to translocate from one side of the NPC to the other multiple times as the FG-nup mesh

continuously changed its conformation (Supplementary Movie 2). The plots of the proteins' CoM $z$ coordinate versus simulation time (Fig. 2d) reveal a telegraph-like behavior, where prolonged intervals of protein diffusion within one of the compartments are interrupted by rapid translocations through the FG-nup mesh to the other compartment. The rate of spontaneous transport from one compartment to the other visibly decreases as the protein mass increases, the proteins are also less likely to approach the NPC's midplane ($z = 0$ nm). For each successful translocation event, there are many more half-way spikes, indicating unsuccessful translocation attempts. Over the aggregate simulation time of 12,000 μs, the largest protein, GAPDH, underwent only one successful translocation and, hence, simulations of larger proteins were not conducted.

We rationalize the protein translocation traces by computing, for each protein species, the distribution of the protein's CoM $z$ coordinate, $P(z)$. Assuming our simulations have adequately sampled the configurational space, we can interpret a Boltzmann inversion of the $z$ coordinate distribution, $-k_B T \ln P(z)$, as a potential of mean force (PMF) acting on the protein as it passes through the NPC. Figure 2e,f shows a representative normalized distribution and the PMF, respectively, for one protein; Supplementary Figs. 4 and 5 show similar data for all protein species. For comparison, protein diffusion simulations were performed using an NPC model that lacked the disordered FG-nup mesh. According to the $P(z)$ and PMF plots, the primary barrier to

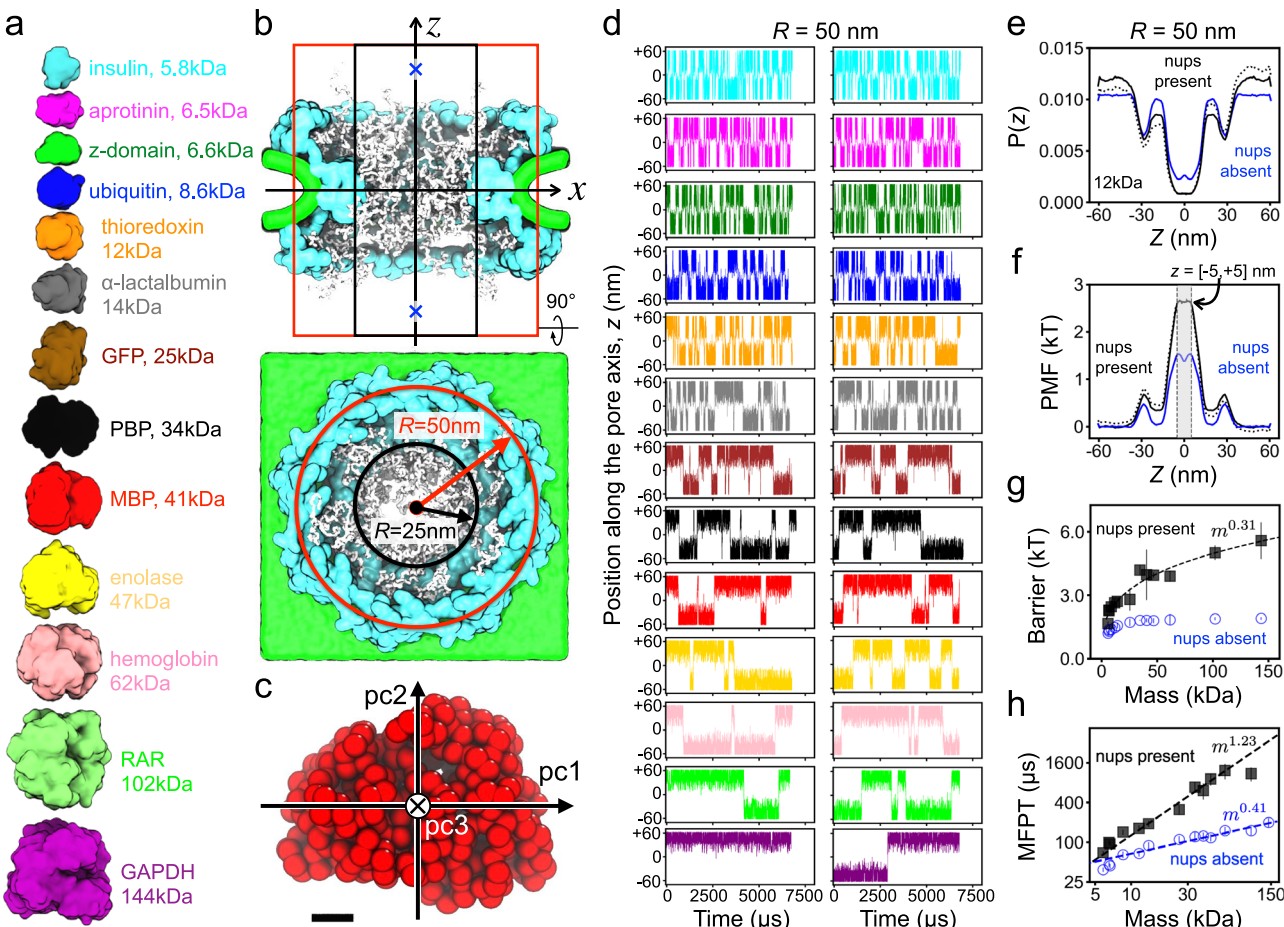

**Fig. 2 | CG simulations of passive diffusion across the NPC. a** To-scale structural representation of all proteins used for CG simulations of passive diffusion. **b** Side (top) and overhead (bottom) view of the simulation systems. Black and red lines indicate the approximate location of a cylindrical confinement potential of 25 and 50 nm radius, respectively. Blue crosses mark the two initial locations of the protein (at $z = \pm 50$ nm). **c** Rigid-body model of a maltose-binding protein (MBP) where each bead represents one amino acid. Pc1, pc2, and pc3 denote the principal axes of the protein. Black scale bar, 1 nm. **d** Center-of-mass $z$ coordinate of each protein (colors defined in panel **a**) versus simulation time. The simulation traces in the two columns differ by the initial placement of the protein. The simulations were performed in the presence of a 50 nm-radius confinement potential. **e** Normalized distribution of the CoM $z$ coordinate of thioredoxin. The black dotted and solid lines show the distribution extracted directly from the simulations and the symmetrized distribution, respectively. The blue line shows a symmetrized distribution for the simulation carried out in the absence of the FG-nup mesh. **f** Potential of mean force (PMF) for thioredoxin transport across the NPC. A PMF barrier is defined as the average value within $|z| < 5$ nm. **g** PMF barrier versus protein molecular mass determined from CG simulations of protein diffusion through our composite Lin2016 NPC model (black squares) and the model devoid of all FG-nups (blue circles). Data presented as mean values ± the average point-by-point difference of the unsymmetrized PMF values from $-50 < z < 0$ nm and $0 < z < 50$ nm intervals. PMF curves derived from $N = 140,000$ frames. $N = 10$ points along those curves defined the mean barrier, and $N = 100$ points defined the average point-by-point difference. Line shows a power-law fit to the data. **h** Mean first-passage time (MFPT) versus protein molecular mass, both axes logarithmic. Power-law fits are shown as dashed lines in panels **g** and **h**. In panel **h** only, the power-law fit to the "nups present" data included proteins up to 62 kDa (hemoglobin).

passive diffusion lies at the NPC's midplane, the location of NPC scaffold's inner ring. The two smaller secondary barriers correspond to the scaffold's outer rings. Accordingly, the secondary peaks are absent in the PMFs extracted from the simulations carried out under a narrower confinement potential (Supplementary Figs. 6 and 7). The amount of asymmetry observed for the raw, non-symmetrized $P(z)$ and PMF$(z)$ curves (Supplementary Figs. 4–7) provides a sense of the sampling quality.

Having defined the PMF barrier as the average value of the PMF profile within 5 nm of the NPC midplane (Fig. 2f), we investigate how the barrier height depends on the protein mass (Fig. 2g). For small proteins (<10 kDa), the protein scaffold is seen to contribute a considerable fraction of the translocation barrier, which can be appreciated from a comparison of the PMFs obtained with and without the FG-nup mesh (Fig. 2f). The effect of the mesh becomes more dominant for larger proteins and the translocation barrier increases with the protein mass, $m$, as $\sim m^{0.31}$. In the case of protein transport through

nup-less NPC, the translocation barrier varies considerably less with the protein mass, as $\sim m^{0.11}$, reflecting a modest change in the available configurational space because of the steric interactions with the NPC scaffold. Conversely, noticeably smaller barriers were extracted from the simulations carried out under a narrower, 25 nm-radius confinement potential (Supplementary Fig. 3) as the scaffold occupied a smaller fraction of the simulation volume.

We characterize the protein translocation time by recording the time elapsed from the moment the protein first enters the NPC volume (defined to be at $z = \pm 20$ nm) to the moment the protein exits the NPC volume on the opposite side, i.e., the first-passage time. Multiple translocations were observed for each protein during our CG simulations (except for GAPDH), and averaging overall translocation events gave the mean first-passage time (MFPT). In the absence of nups, the MFPT is observed to increase with the protein mass as $m^{0.41}$ (Fig. 2h) or as $m^{0.44}$ (Supplementary Fig. 3f), depending on the width of the confinement potential. This scaling exponent is close to but slightly higher

than the free diffusion limit ($m^{1/3}$), which we attribute to small yet not negligible effect of the NPC scaffold. A much steeper dependence was observed for the Lin2016 NPC model with FG-nups: the MFPT increasing as $m^{1.23}$ (Fig. 2h) or as $m^{0.97}$ (Supplementary Fig. 3f). Thus, the transport rate in the absence of FG-nups appears to scale, approximately, with the protein radius (~$m^{1/3}$), and, in the presence of nups, with the protein volume (~$m^1$) for the range of masses explored by our CG simulations. Test simulations featuring two proteins (ubiquitin and GFP) diffusing simultaneously through the NPC (Supplementary Fig. 8), yielded MFPT values ($128 \pm 20\,\mu s$, ubiquitin; $353 \pm 181\,\mu s$, GFP) within one standard deviation of the values obtained from our single-protein simulations ($142 \pm 12\,\mu s$, ubiquitin; $312 \pm 52\,\mu s$, GFP).

An experimental study that measured the accumulation of dye-labeled GFP dimer (about 50 kDa) into the nucleus of permeabilized HeLa cells[54] provides us the opportunity to compare our simulated transport rates to experiment. Under a 50 nm-radius confinement, the protein concentration in our simulations is about $5\,\mu M$. Linearly extrapolating from the experimentally measured transport rate to a GFP dimer concentration of $5\,\mu M$, the expected experimental transport rate is ~300 molecules/s for one NPC, which corresponds to a mean first-passage time of about 3.3 ms. In our CG simulations, mean first-passage time of a protein of a comparable size (enolase, 47 kDa) is about 0.9 ms (Fig. 2h), which is a factor of four faster than expected from experiment. Thus, the simulated timescale of the NPC transport is of the same order as in the experiment.

**Free diffusion determines the successful crossing timescale**

The telegraph-like shape of the protein permeation traces (Fig. 2d and Supplementary Fig. 3b) suggests that the timescale of a successful crossing event is orders of magnitude smaller than the timescale separating successful crossings. The latter is expected to depend on

the effective concentration of the protein and, indeed, we observe about a three-fold decrease in the number of crossing events when changing the radius of the confinement potential from 25 to 50 nm, slightly less than the expected factor of four because a part of the additional volume of the larger confinement potential is occupied by the NPC's scaffold. Figure 3a and Supplementary Movie 3 illustrate one representative crossing of a maltose-binding protein (MBP), which completes in about 6 µs. The passage of the MBP along the pore axis is accompanied by a similar magnitude displacement perpendicular to the pore axis. The configuration of the FG-nup mesh rearranges significantly over the course of the crossing event. The MBP permeation trace (Fig. 3b and c) shows that the MBP's position during the crossing, i.e., from 5331 to 5337 µs, decreases almost monotonically in $z$, after a partial (failed) crossing immediately beforehand. A zoomed-in view on another crossing event (Fig. 3d) from the same simulation shows significant back-tracking, but, ultimately, the protein crosses the NPC in less time than in the event shown in Fig. 3c. A careful look at the simulation trace (Fig. 3b) reveals that when a protein enters the NPC volume (defined as $|z| < 20$ nm), the protein almost always fails to complete the crossing, although the relative rate of success and failure depends on the protein size.

To characterize the timescale of the crossing events, we define a crossing time as the time elapsed from the moment the protein exits the NPC volume (at $z = \pm 20$ nm) to the last prior moment the proteins crossed the NPC boundary on the other side of the NPC, see Fig. 3c,d for two examples. Note that this definition differs from that of a first passage, as the crossing time does not account for the time the protein spends meandering on the entrance side of the NPC volume after crossing its boundary for the first time. Using the above definition of the crossing time, we collected crossing time statistics for six proteins ranging in their molecular mass from 25 to 102 kDa. The average

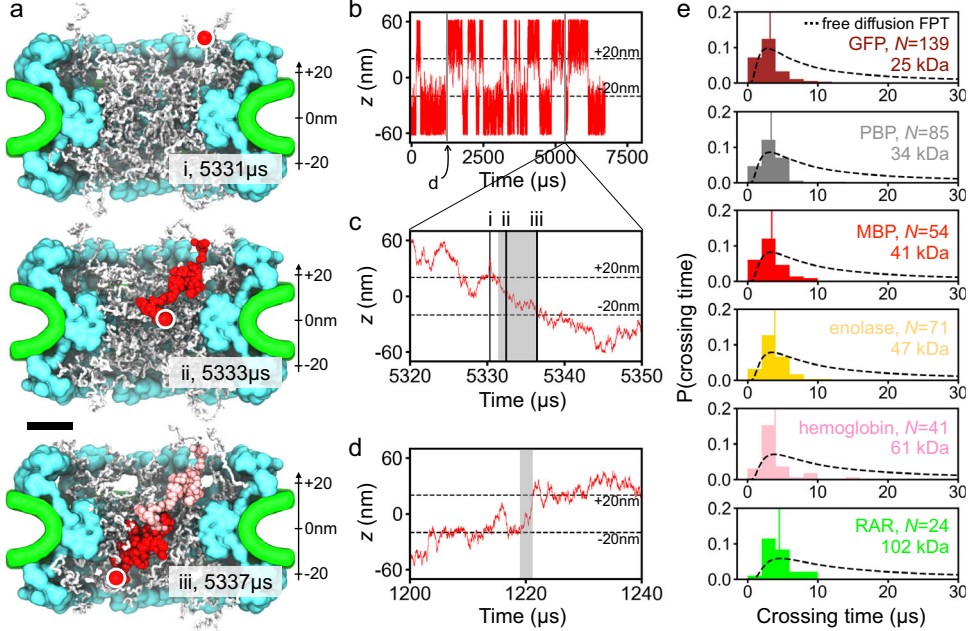

**Fig. 3 | Timescale of protein crossings. a** Successful translocation of a maltose-binding protein (MBP, red) through the NPC. Three instantaneous configurations of the NPC are shown using white for the FG-nups, cyan for the protein scaffold and green for the lipid bilayer. The circle indicates the location of the MBP protein at each configuration. Red and pink beads illustrates the configurations explored by the protein between the instantaneous configurations (sampled every 25 ns). Scale bar, 20 nm. **b** CoM $z$ coordinate of MBP simulated under a 25 nm-radius confinement potential. Two crossing events form this trace are shown in detail in panels **c** and **d**. **c** Zoomed-in on the crossing event trace. The same event is illustrated by

snapshots (i, ii, iii) in **a**. **d** Example of another crossing event. The gray rectangles in panels **c** and **d** illustrate the time interval defined as a crossing time for the analysis shown in **e**. **e** Distribution of crossing times for six protein species. $N$ specifies the total number of crossing events used to construct each histogram. Each normalized histogram was constructed using 15 evenly-spaced bins, from 0 to 30 µs, and the crossing time data from both confinement potential simulations. The average of each histogram is shown as a vertical line. The dashed lines show the distributions of first-passage time for a freely diffusing particle of the same diffusion constant as that of the corresponding protein.

crossing time is seen to shift towards the right and, although its molecular mass dependence is considerably less pronounced than that of the MFPT (Fig. 2h and Supplementary Fig. 3f), it does not depend on the width of the confinement potential and appears to roughly follow the dependence of the diffusion constant on the protein mass (average crossing time ~ $m^{0.26}$ whereas $1/D \sim m^{1/3}$). For reference, we plot in Fig. 3e theoretical distributions, $P(t) = (l/\sqrt{4\pi D t^3}) \exp(-l^2/4Dt)$, of the time, $t$, required for a particle of the same diffusion constant as that of the corresponding protein to travel distance $l = 40$ nm along a straight line via free diffusion under open and absorbing boundary conditions at the beginning and the end states of the process. We conclude that the average successful crossing time is close to but even faster than the mean first-passage time of a freely diffusing particle, due in part to including time for the free particle's failures to translocate across.

Thus, our analysis of simulation trajectories suggests that the process of passive transport through the NPC consists of many unsuccessful translocation attempts interrupted by infrequent successful translocation. The timescale of successful translocations, however, is found to be prescribed by the timescale of protein-free diffusion. We interpret these observations as a process where constantly rearranging FG-nups open and close transient passages that a protein can take to cross from one side of the NPC to the other at speeds prescribed by free diffusion. Note that this scenario contrasts with a situation where the action of FG-nups is to slow down the effective diffusion of the proteins, through either entanglement or binding, which would manifest itself in a slower than free diffusion timescale of successful crossings.

## The barrier to passive transport as a percolation transition

We prove our open passage model of passive transport by developing a theoretical approach that can predict the rate of passive transport of proteins through the NPC from the analysis of equilibrium fluctuations of the FG-nup mesh alone. For each instantaneous configuration of the mesh sampled by the equilibration trajectories (Fig. 2), and for each spherical probe of radius $R_p$ (Supplementary Table 2), we classified each voxel within the volume of the NPC as available if the sphere could be placed at that voxel without clashes with either the mesh, the scaffold, the envelope or the confinement potential, producing a 3D map of internal voids within the NPC volume (Fig. 4a), see methods for a detailed description of the procedure. The 3D void map was converted into a 1D potential occupancy map, $P_1(z)$, by splitting the NPC volume into disc segments along the pore axis and calculating the fraction of available voxels in each segment (Fig. 4b). Trajectory average of the potential occupancy function (Fig. 4c) was converted into an effective PMF (Fig. 4d), through Boltzmann inversion. Gratifyingly, the shape of the resulting PMF reproduces the shape of the PMF extracted directly from brute-force simulations (Fig. 2f).

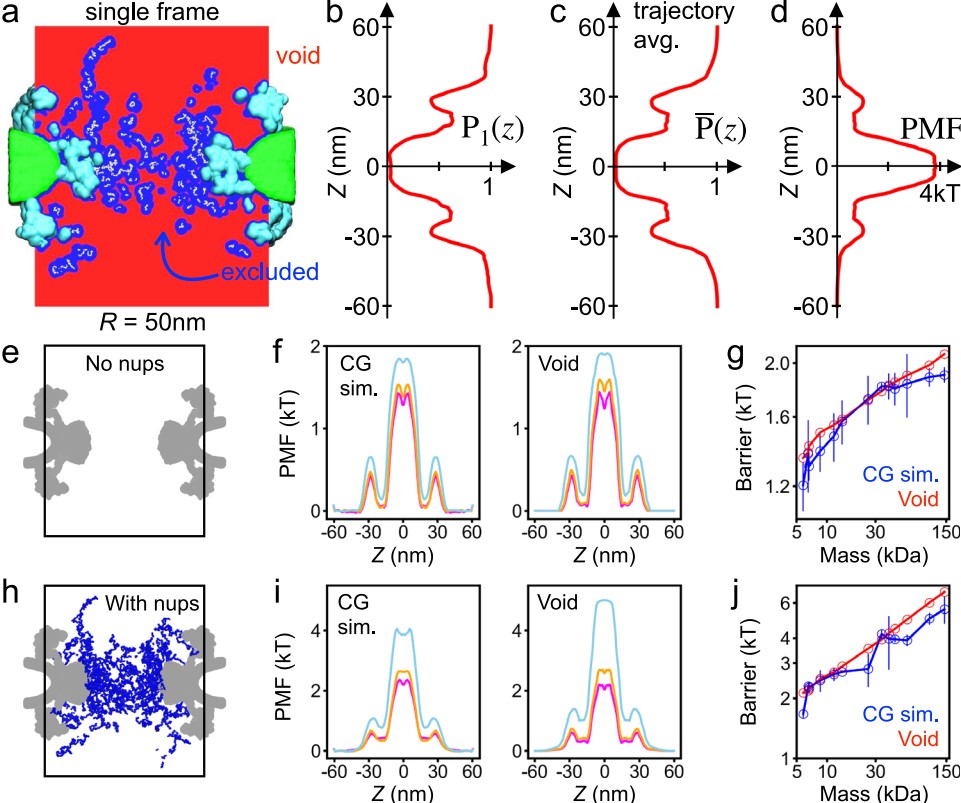

**Fig. 4 | Void model of the translocation barrier. a** Void analysis map of an instantaneous NPC configuration computed using a spherical probe of 22.4 Å radius. The volume available to accommodate the probe (void) is shown in red, the volume excluded in blue, FG-nups in white, the scaffold in cyan and the lipid bilayer in green. The image shows a 2D section of a 3D map. **b** The fraction of the NPC volume that can accommodate the probe without clashes as a function of the pore axis coordinate. The fraction was computed by splitting the void analysis map into cylindrical segments of 50 nm radius and 0.6 nm height, co-axial with the pore. The data shown were computed for the instantaneous NPC configuration displayed in panel **a**. **c** Trajectory-averaged probability of accommodating the probe as a function of the pore axis coordinate, $\overline{P}(z)$, computed by averaging instantaneous void analysis maps over the last 6 ms of the NPC equilibration trajectory, sampled every 1.0 μs. **d** PMF of the spherical probe derived by void analysis. **e** Cut-away view of the NPC system containing no nups, only the scaffold and nuclear envelope potentials (gray). **f** Symmetrized PMF of three protein species (aprotinin, magenta; thioredoxin, orange; and hemoglobin, light blue) derived from brute-force CG simulations (left) and of the three spherical probes of approximately the same radius ($R_p = 12.75$, 15.71, and 30.04 Å) derived from void analysis (right). **g** PMF barrier versus protein mass. Interpolation was used to find void analysis PMF barriers for the proteins simulated using the CG method (Supplementary Fig. 10). Lines are guides to the eye. Both axes use logarithmic scale. **h–j** Same as in **e–g** but for the NPC model that includes nups.

Further comparison shows that our void analysis method quantitatively reproduces the proteins' PMF. To make such a comparison possible, we used interpolation to convert the radius of a spherical probe to the corresponding protein mass (Supplementary Fig. 9), and to find the height of the PMF barrier (Supplementary Fig. 10). In the case of a nup-less NPC (Fig. 4e), the method reproduced fine features of the PMF, including a small dip near the pore midplane, i.e., at $z = 0$ (Fig. 4f), which is caused by a small widening of the NPC scaffold at the very center of the inner ring. The method also reproduces the absolute height of the barrier (Fig. 4g) without any adjustable parameters. The good quantitative agreement was also observed for the Lin2016 NPC model with FG-nups (Fig. 4h–j) and also for simulations carried out under a narrow confinement potential (Supplementary Fig. 11). We note that, for two trajectories of the same length, the data derived from the void analysis method have better statistical sampling because the method characterizes potential occupancy of the entire simulation volume, whereas sampling in a CG simulation is limited to the location of the diffusing protein. The void analysis method also provides a means to estimate the free-energy barrier to passive diffusion of proteins that are too large to obtain good passage statistics from brute-force simulations.

Matching the definition of the first-passage used to characterize protein permeation in our CG simulations (Fig. 2h), we compute MFPT by numerically solving the Fokker-Planck equation using our void analysis PMFs and position-dependent diffusion constants, see Methods for a detailed description of the procedures. Supplementary Fig. 12 shows the obtained dependence of the MFPT on the void probe radius as well as the interpolation scheme used for direct comparison of the data with the results of the CG simulations. For our Lin2016 NPC model, both with and without FG-nups present, the MFPTs computed using our Fokker-Planck approach are in excellent agreement with the CG simulation data for both wide (Fig. 5a, b), and narrow (Supplementary Fig. 13a, b), confinement potentials, as well as with the simulation and experimental results of previous studies[6,8], when scaled by the MFPT of hemoglobin (Supplementary Fig. 14). Note that, in comparison to the Fokker-Planck approach, the CG simulations are expected to systematically underestimate the MFPT of larger proteins because the MFPT distributions are expected to have long tails that are not sampled by the limited-duration CG simulations.

In the absence of nups, both approaches yield nearly identical dependences of the MFPT on the protein mass, with a single power-law spanning the entire range of protein masses (Fig. 5c and Supplementary Fig. 13c). In the presence of FG-nups, the dependence no longer could be described by a single power law (Fig. 5d and Supplementary Fig. 13d). First, we note that even the largest proteins (11–13 nm in diameter) examined above are considerably smaller than the narrowest cross-section (44 nm in diameter) of the NPC scaffold, although the latter can vary among biological species[41], the development stage of an organism[43] or even tension[42]. For larger proteins, we indeed observe the nuclear pore to serve as a barrier to passive diffusion, slowing down spontaneous translocation of an 11 nm-diameter protein by a factor of 1000 compared to an empty pore.

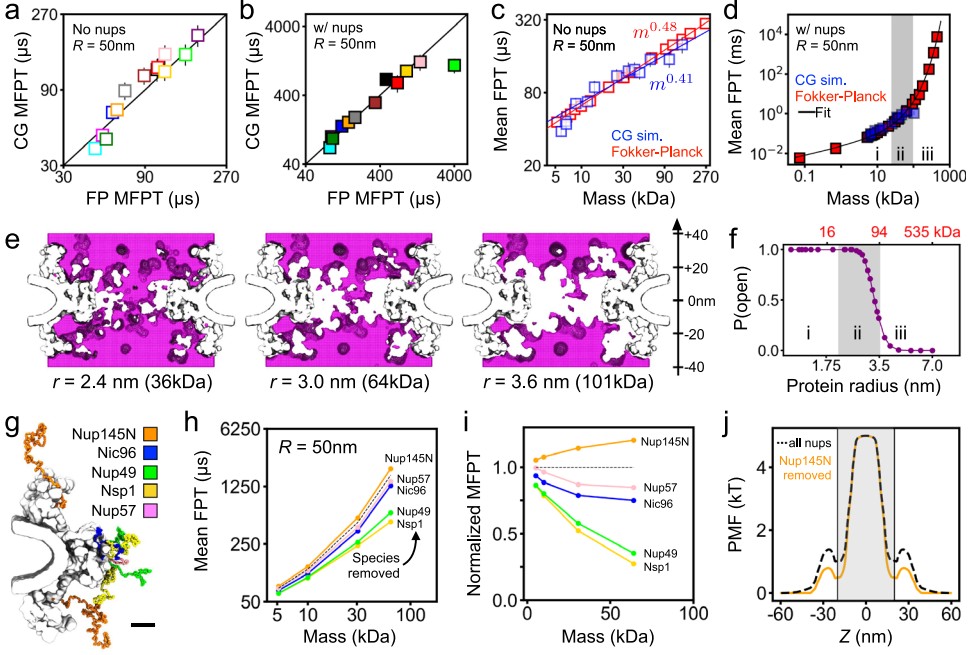

**Fig. 5 | Fokker-Planck model of passive diffusion and the percolation transition. a, b** Comparison of the mean first-passage times (MFPT) calculated from CG simulations to those from our Fokker-Planck void analysis model for the Lin2016 structure without FG-nups (**a**) and with FG-nups present (**b**). The black line indicates perfect agreement. Data in **a, b** presented as mean values ± SEM, based on traces involving $N = 140,000$ frames. **c** MFPT from CG simulations (blue) and using our Fokker-Planck approach (red) as a function of protein molecular mass. Note the logarithmic scale of the axes. Power law fits, and their slopes, specified in the figure. **d** Same as in panel c but for the Lin2016 structure with FG-nups present. The black line shows a single fit to all of Fokker-Planck data, see text for the functional form. The three regions (i, ii, iii) correspond to power law, transition and exponential scaling behavior. **e** Connectivity map of an instantaneous NPC configuration computed for three different protein probe radii (specified under each map). Each map generated for $|z| < 40$ nm.

**f** Probability of finding a complete, open path through the NPC versus protein radius and molecular mass (red, top) obtained from the analysis of NPC equilibration trajectories. Note the logarithmic scale of the horizontal axes. The three regions (i, ii, iii) correspond to those in panel **d**. **g** Location of each FG-nup species in one sixteenth of the CG model. Scale bar, 10 nm. **h** MFPT versus molecular mass for an NPC model devoid of one FG-nup species (colors) and with all FG-nups present (dashed black line). **i** MFPT for the deletion mutants normalized by the MFPT of the complete NPC model. **j** PMF of a 64 kDa ($r = 3.0$ nm) protein for all species present NPC (dashed black line) and the Nup145N deletion mutant (orange). The gray shaded region ($|z| < 20$ nm) indicates the region used to define the pore length of the central channel. All data in this figure were obtained under a 50 nm-radius confinement potential. Interpolation was used to express the results of the Fokker-Planck void analysis model in terms of molecular mass (Supplementary Fig. 12).

To systematically investigate the crossover behavior, we fitted the dependence of the MFPT on the protein radius using the following expression:

$$\tau(R_p) = \tau_0 R_p \exp\left(\frac{R_p + 3A}{R_0}\right)^{\alpha}, \tag{1}$$

which derives from the transition rate expression $\tau \sim e^{H_{eff}}/D_{eff}$, where $H_{eff}$ is the effective barrier height and $D_{eff}$ is the effective diffusion coefficient. According to the Stokes–Einstein equation, the effective diffusion coefficient scales as $R_p^{-1}$, which leads to the $\tau_0 R_p$ pre-exponential factor, where $\tau_0$ is a constant proportional to protein concentration. In Eq. (1), we assume the following scaling relation for effective barrier height: $H_{eff} = \left(\frac{R_p + 3A}{R_0}\right)^{\alpha}$, where the 3 Å term is added to the protein radius to take into account the physical size of the FG-nup residues, $R_0$ is the threshold radius, and the scaling exponent is $\alpha$. The fitting yields $\tau_0 = 3.26\ \mu s/\text{Å}$, $R_0 = 19.8\ \text{Å}$ and $\alpha = 1.89$ for the 50 nm-radius confinement, and $\tau_0 = 1.54\ \mu s/\text{Å}$, $R_0 = 22.5\ \text{Å}$ and $\alpha = 2.01$ for the 25 nm confinement. When $R_p < R_0$, the relation between MFPT and $R_p$ resembles a power law since the exponential functions could be approximated by two-term Taylor series. The exponential term dominates when $\ln(R_p)$ becomes smaller than $((R_p + 3A)/R_0)^{\alpha}$, which, in the case of the 50 nm confinement, corresponds to $R_p > 35.9\ \text{Å}$ and to $R_p > 39.26\ \text{Å}$ for the 25 nm one. Accordingly, we can divide the dependence of the MPFT on protein mass (Fig. 5d) into power law, transition and exponential scaling regions, which suggests a transition from a soft to hard translocation barrier with increasing protein mass. Note that importin-$\beta$ (95 kDa), a representative karyopherin (i.e., nuclear transport) receptor, by its size alone, is expected to experience a considerable reduction of spontaneous transport because of the presence of the FG-nup mesh.

To determine the physical origin of the crossover behavior, we analyzed connectivity of the transient voids formed within the FG-nup mesh. For each instantaneous mesh configuration, we search the 3D void map for a path crossing the NPC volume ($|z| < 20$) that is accessible to a protein of a given radius. To illustrate the procedure, Fig. 5c shows cross sections of three void maps computed for the same mesh configuration and three proteins of different effective radii, i.e., 2.4, 3.0, and 3.6 nm. Upon increasing the protein radius, the path connecting the two sides of the NPC becomes fragmented as some voxels near the inner ring of the NPC scaffold become inaccessible. We estimated the instantaneous probability to have an open path for a protein of a particular size by performing the above analysis on more than 8000 FG-nups configurations obtained from the equilibration CG simulations of the NPC model (Fig. 1a). The resulting probability function (Fig. 5f) displays a transition from always having at least one fully connected path to not having a path at all most of the time, a percolation transition. Interestingly, the percolation transition (Fig. 5f) occurs at the same protein mass as where the dependence of the MFPT on protein mass changes from power law to exponential (Fig. 5d).

We interpret the results of our analysis in a physical model of passive transport where, for small proteins, an open path through an NPC always exists, and the protein diffusion is limited by the likelihood of finding this path by diffusion. For larger proteins, the rate of passive transport is also conditioned by a small yet finite probability of forming an open path through the FG-nups as a transient fluctuation, resulting in a stronger barrier to translocation that sharply increases with protein mass.

Using our theoretical model, we determined the relative contribution of each FG-nup species to the diffusion barrier. Five additional equilibration trajectories were generated by computationally removing all residues of each of the five FG-nup species (Fig. 5g). The resulting trajectories were analyzed using our theoretical model, yielding the dependence of the MFPT on the protein mass for each deletion version of the NPC (Fig. 5h). Normalized by the MFPT

observed for the complete NPC model, we find Nsp1 and Nup49 species to contribute the most to the diffusion barrier (Fig. 5i), which could have been expected given their length and tethering position. The small increase of the MFPT produced by the removal of Nup145N is explained by the small change in the PMF, which deepened the PMF minima between the inner and outer rings of the scaffold (near $z = \pm 20$ nm) without affecting the height of the primary barrier (Fig. 5j). As the location of those PMF minima coincide with the span of the computational domain used to solve the Fokker-Planck equation, removal of Nup145N effectively made the translocation barrier larger, increasing the MFPT.

**Passive transport through yeast NPC**

To illustrate the robustness of our findings, we performed CG simulations and void analysis procedures on another experimentally derived NPC structure[55], which we refer to as hereafter as the Kim2018 structure. Starting with the Kim2018 scaffold and with the known stoichiometry of the yeast NPC, we built a CG model of the Kim2018 structure and equilibrated it for 7.5 ms using the same protocols as in the case of the Lin2016 structures, see Methods for details. A representative equilibrated configuration of the Kim2018 model is shown in Fig. 6a. In accordance with the composition of the Kim2018 structure (which lacks symmetry with respect to the NPC's midplane), the resulting average density of the FG-nups has an asymmetric shape (Fig. 6b), but spans a similar range of density values as in the Lin2016 model (Fig. 1b). The average density of the FG-nup amino acids within the central region of the mesh (a cylinder of 30 nm diameter and 30 nm length centered at the origin) was $68.8 \pm 2.7$ mg/mL for the Kim2018 model, slightly higher than for the Lin2016 model, $59.8 \pm 3.0$ mg/mL, where the standard deviation reflects temporal fluctuations when sampled every 0.1 μs. We attribute that difference in density, in large part, to the fact that the Kim2018 model contains 48 copies from Nup100+116+145N, whereas the Lin2016 NPC model only contains 32 copies of Nup145N (anchored more externally), see Supplementary Fig. 1 for the stoichiometry of each structure.

Having sampled the ensemble of FG-nup conformations for the Kim2018 model, we applied our void analysis method to determine the PMF for protein passage across the NPC. The resulting PMF profiles (Fig. 6c) are slightly asymmetric and contain a single peak, unlike the three-peak PMF profiles observed for the Lin2016 model (Fig. 4i), which we attribute to the presence of large outer rings in the Lin2016 scaffold, absent in the Kim2018 scaffold. According to our PMF calculations (Fig. 6d), a protein of the same size experiences a slightly greater PMF barrier for the passage across the NPC in the Kim2018 structures compared to Lin2016, which we attribute to the higher overall FG-nup density of the former structure. Similarly, the MFPT values computed using our Fokker-Planck approach for the Kim2018 model were found to be higher than those obtained for the Lin2016 structure (Fig. 6e). The dependence of the MFPT on protein size is seen to exhibit a soft-to-hard barrier transition well-approximated by the same functional form, Eq. (1), as in the case of the Lin2016 model. Importantly, the Kim2018 model was found to exhibit a percolation transition as a function of the probe radius (Fig. 6f), congruent with the power law-to-exponential change in the scaling behavior.

We built another coarse-grained representation of the yeast NPC based on the integrative scaffold constructed by Kim et al.[55], inspired, in part, by the composition of a yeast NPC model simulated by Huang et al.[35], which we refer to as "Huang2020" (Supplementary Fig. 15a–c). Compared to our Kim2018 model, two new FG-nup species were included—Nup42, Nup2—and flexible linker regions restrained only on their terminal ends for Nup116, Nup100, Nup145N in the version we call "Kim2018+." In all, the FG-nup domains of Kim2018+ have about 30% more amino-acid residues present than in our Kim2018 model, and the overall FG-nup mass of

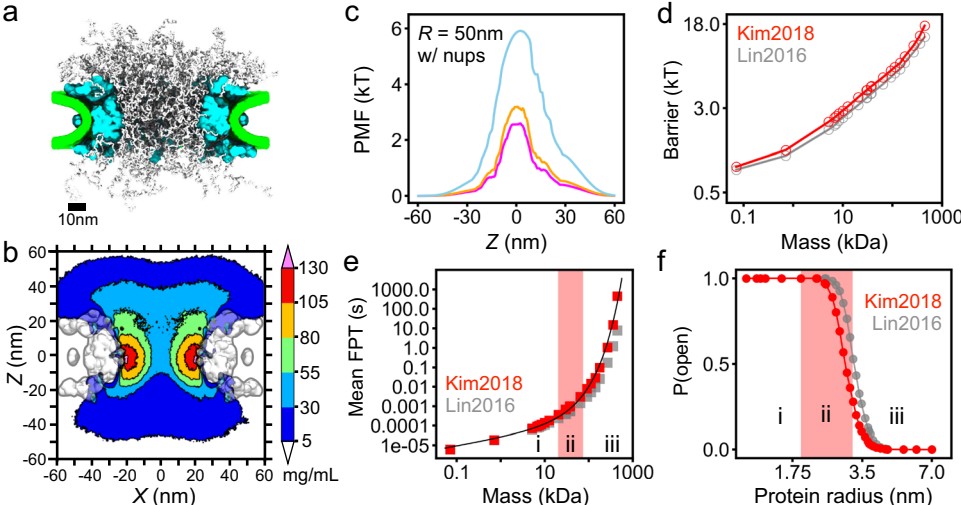

**Fig. 6 | Passive diffusion of proteins through yeast NPC. a** Representative configuration of the CG model of the integrative Kim2018 structure of yeast NPC[55]. The FG-nups are shown in white, protein scaffold in cyan and lipid bilayer in green. **b** Cross-section of the average FG-nup amino-acid density. The 3D density map was generated by averaging instantaneous configurations of the Kim2018 computational model every 0.1 μs over the final 6 ms fragment of the 7.5 ms simulation. The cross-section was averaged along the *y* coordinate within the (−7.5, +7.5) nm range. **c** Potentials of mean force along the pore axis of the Kim2018 model, calculated by void analysis for probe radii of 12.75, 15.71, and 30.04 Å (magenta, orange, and light blue, respectively). **d** PMF barrier as a function of protein mass, calculated by void

analysis for the Kim2018 (red) and Lin2016 (gray) models. **e** Mean first-passage time versus protein mass for the Kim2018 (red) and Lin2016 (gray) models. The black line is a single fit to all of the Kim2018 data, the functional form is provided in the text. The fitting yields $\tau_0 = 3.8$ μs/Å, $R_0 = 19.0$ Å, and $\alpha = 2.05$. **f** Probability of finding a complete, open path through the Kim2018 model of the NPC versus protein radius. The three regions (i, ii, iii) correspond to power law, transition and exponential scaling behavior. To enable direct quantitative comparison of the Kim2018 and Lin2016 models, we defined the central channel to span the $|z| < 20$ nm region in both models. All data reported in this figure were obtained under a 50 nm-radius confinement potential.

Kim2018+ is quite close to that of Huang2020. The exact compositions of these models (Kim2018, Kim2018+, and Huang2020) are provided in Supplementary Table 3. From Supplementary Fig. 15d–f, we observe that the additional FG-nup domains in Kim2018+ lead to a slightly increased density within the central channel and on the nucleoplasmic side (i.e., at negative values of *z*). The difference in density, however, does not exceed 30 mg/mL, and it is less than 5 mg/mL in several regions of the NPC volume (Supplementary Fig. 15f). We anticipate, therefore, that the PMF barrier heights would be slightly higher, and MFPT slightly longer, for Kim2018+ compared to Kim2018. Furthermore, we expect the percolation transition that corresponds to a power law-to-exponential change in MFPT scaling behavior would occur at a slightly smaller protein size for Kim2018+ compared to Kim2018. We expect the differences would be relatively small between these two models, however.

Thus, we have found a percolation transition to prescribe a size limit for passive transport through two models of the NPC built using two independent experimental structures. Although the precise value of the percolation transition is found to depend on the overall density of the FG-nup mesh, the physical mechanism enabling protein size selection remained unchanged, and hence we believe it to be a general feature of a disordered mesh. For NPC structures featuring wider scaffolds but similar FG-nup stoichiometry[41,42,56], we expect the percolation threshold to shift toward larger protein sizes compared to the values determined for the Lin2016 and Kim2018 structures. Conversely, the presence of 50 nuclear transport factors, such as importin-β, within the central mesh of the NPC[57] is expected to increase the effective FG-nup density by ~11%, pushing the percolation threshold (and hence the soft-to-hard barrier transition) to smaller protein sizes. Similar shifts in the percolation transition can be expected for the NPC scaffolds that can change their size over time, as seen in the AFM studies[58]. As a reference, we note that a ~10% change in the scaffold ring diameter would amount to a ~20% change of the FG-nup density, a density change of the same magnitude as the difference between the Lin2016 and Kim2018 structures.

## Discussion

We have constructed structurally accurate computational models of two NPC systems and carried out CG simulations to find the passive transport through the NPC to be dominated by rare, fast crossings. We have shown that such crossings occur in a free diffusion regime, which suggests a mechanism whereby the protein transport through the mesh is conditioned by the presence of open paths connecting one side of the NPC to the other. We have shown that to be the case by constructing a theoretical model of passive transport derived from geometrical analysis of the volume available to accommodate the translocating molecule. In that respect, our model is similar to the approach used by Bodrenko and co-workers to describe the passive diffusion of antibiotics through a membrane channel[59]. Based solely on the analysis of the free volume, our theoretical model could not only reproduce the transport rates observed in our CG simulations but also allowed us to estimate the transport rates for the proteins that were too large to be characterized through brute-force simulations. We find that the average translocation time initially increases proportionally to the protein molecular mass but undergoes a crossover to an exponential dependence for larger masses. By analyzing how the connectivity of the two compartments depends on the protein mass, we associate the crossover in the scaling behavior with a percolation transition, which indicates a qualitative change in the character of the passive transport from being dominated by the protein finding an entrance to a connecting path to the proteins waiting for the connecting path to form. This change in the rate-limiting step enables the NPC to function as a protein size filter presenting a soft barrier to diffusion of small proteins and a hard barrier to the transport of larger proteins, with the protein size cutoff being determined by the percolation transition.

Our void model of passive transport has several limitations. The model is designed to work for globular proteins that do not exhibit specific binding to FG-nups, which would be the case for assisted nuclear transport. It is, however, conceivable that introducing one or more binding sites along a quasi-1D path through the NPC mesh could increase the rate of transport in comparison to the passive diffusion

case, as suggested by previous theoretical work[22–25]. Although the model assumes the proteins to be spherical, non-spherical shapes can be accommodated in the model at the level of the PMF calculations by modifying the void search protocols[60]. The model is not expected to work for partially or fully disordered proteins or other polymer-like molecules (such as mRNA) because polymer entanglements are not accounted for in our Fokker-Planck formulation of the transport. The void analysis approach could be extended to describe a situation where multiple protein species diffuse through the mesh by iterating the void finding the procedure with a probabilistic placement of proteins within the voids according to the proteins' PMFs and bulk concentrations until convergent PMF profiles are established for all protein species.

Our experimentally derived CG models set the stage for the further computational exploration of nuclear transport. We envision applying our model to study passive transport through alternative models of the NPC structure[41,42] as well as incorporating specific binding in the model to study assisted transport of larger cargoes. Accounting for the chemical reactions that give nuclear transport its directionality and for the recycling of the transport factors would furnish the first complete physics-based model of nuclear transport.

## Methods

### Overview of CG simulation methods

In our CG models of the NPC, the disordered mesh region is represented using a beads-on-a-string model previously developed by the Onck lab[48,49], whereas the nuclear envelope and the structured protein scaffold are represented using custom grid-based potentials. The proteins diffusing through the NPC are described as rigid bodies. Each component of the model is described in detail below.

### CG model of a composite NPC structure

In accordance with the stoichiometry of the composite NPC structure[45], which we will refer to as "Lin2016," the disordered mesh regions consisted of 32 copies of each of the following five species: Nup145N, Nic96, Nsp1, Nup49, Nup57, 160 individual protein chains in total. The amino-acid sequence of the disordered domain of each protein species was determined by comparing the full-length protein sequence (UniProt entry G0SAK3 for Nup145N, G0S024 for Nic96, P14907 for Nsp1, G0S4X2 for Nup49 and G0S0R2 for Nup57) to the corresponding sequence of the protein domain resolved in the structure of the protein scaffold[45]. Thus, the disordered mesh region consisted of Nup145N (residues 1–732), Nic96 (1–139), Nsp1 (1–467), Nup49 (1–245), Nup57 (1–77), each being a continuous N-terminal fragment of the corresponding full-length protein. Note, that the unresolved portions of other adaptor nups and coat nups present in the Lin2016 structure are either helical or unstructured linkers, which are proposed to primarily link the nups together within the scaffold. Ref. 45 classifies Nup159 as a cytoplasmic filament nup and does not include the short resolved portion in the published composite symmetric core. Therefore, although Nup159 contains an FG-repeat section, we chose to exclude that nup species from our CG model.

Each amino-acid residue of the fragments was represented by one CG bead; the beads were connected into a polymer chain via harmonic spring potentials describing the bond, angle, and dihedral angle terms, see refs. 48 and 49 for details. We validated our implementation of the CG model by measuring the Stokes radii of three FG-nup species in isolation and the maximum height of a $10 \times 10$ Nup62 brush (Supplementary Fig. 16), which reproduced the values reported in ref. 49. The C-terminal bead of each fragment was harmonically restrained ($k = 10.0$ kcal mol$^{-1}$ Å$^{-2}$) to the location prescribed by the composite NPC structure[45]. The initial configuration of each fragment was generated as a self-avoiding random walk, with the distance between consecutive beads of 0.38 nm and under conditions that no beads of separate fragments were within 0.8 nm of each other and no beads at least three residues away from the anchor point were within 0.8 nm of

the protein scaffold. Two complete sets of 160 FG-nup fragments were generated independently, providing initial conditions for the two replica simulations that utilized the same model of the nuclear envelope and the protein scaffold.

The nuclear envelope was represented using a purely steric, repulsive potential. The overall shape of the potential was obtained by manually fitting a surface to the center line of the apparent lipid bilayer density seen to surround the NPC in the cryo-ET structure, EMD-3103[46], resulting in the following parametric equation for the surface:

$$(r - 0.6B)^2 + 4z^2 = B^2, \qquad \text{if } |z| < 97.2,$$

$$\left[r - 0.6B - (|z| - 97.2)^2\right]^2 + 4z^2 = B^2, \qquad \text{otherwise,}$$

where $z$ and $r$ are the pore axis and radial coordinate, respectively, in Å, and $B = 300$ Å. The first equation above represents an ellipse, translated away from the origin and rotated about the pore axis. The second equation effectively flattens the curvature of the ellipse so that the lipid bilayer is not raised in the corners of the system. Using a custom Python script, the surface was converted to a triangular mesh consisting of 1266 faces and an average area of individual triangles of ~25 nm$^2$. The triangular mesh was provided to the `lipidwrapper` tool[61] along with a square (20 nm on side) patch of a diphytanoyl phosphatidylcholine (DPhPC) membrane, which was replicated and transformed to construct the curved nuclear envelope without gaps. The resulting all-atom model was cut along the $x$ and $y$ axes at ±60 nm. The all-atom representation was converted to a $120 \times 120 \times 76$ nm$^3$ number density map (in units of atoms/Å$^3$) at 0.4 nm resolution using the `volmap` tool of VMD[62]; a spherical gaussian blur was applied to each atom with a standard deviation of eight times the nominal atomic radius. The 3D density map was applied as a potential to the FG-nup beads through linear interpolation, producing a potential that ranged in values from 0 to 15 kcal/mol.

The structured protein scaffold was represented by 20 grid-based potentials, each describing the interaction of an individual amino-acid of a particular type with all amino acids of the scaffold. The potentials were derived by first converting the composite all-atom model of the NPC protein scaffold[45] to a one-bead-per-residue representation. Following that, the number density of type Y residues in the protein scaffold, $\rho_Y$, was obtained using the `volmap` plugin of VMD and with each bead being represented using a normalized gaussian distribution of 1.5 Å width. The scaffold potential for residue type X at position $\vec{r}$ was computed as

$$V_X(\vec{r}) = \sum_Y \int d\vec{r}' K_{XY}(\vec{r} - \vec{r}')\rho_Y(\vec{r}'), \tag{2}$$

where $K_{XY}(\vec{r} - \vec{r}')$ is the potential between beads of type X and Y located at $\vec{r}$ and $\vec{r}'$, respectively. Eq. (2) was integrated numerically using a Fast Fourier transform-based convolution algorithm using a cubic grid of 0.6 nm resolution.

### CG models of a yeast NPC

In accordance with the stoichiometry of an integrative structure of a yeast NPC[55], which we will refer to as "Kim2018," the disordered mesh regions consisted of 48 copies of Nsp1, 16 copies of Nup159, 16 copies of Nup116, 16 copies of Nup100, 32 copies of Nup49, 32 copies of Nup57, 16 copies of Nup145N, 8 copies of Nup1, and 16 copies of Nup60, 200 individual protein chains in total. The amino-acid sequence of the disordered domain of each protein species was determined by comparing the full-length protein sequence (UniProt entry P14907 for Nsp1, P40477 for Nup159, Q02630 for Nup116, Q02629 for Nup100, Q02199 for Nup49, P48837 for Nup57, P49687 for Nup145N, P20676 for Nup1, and P39705 for Nup60) to the corresponding sequence of the protein domain resolved in the structure of the protein scaffold[55]. The disordered mesh region consisted of Nsp1

(residues 1–620), Nup159 (1–760), Nup116 (1–760), Nup100 (1–560), Nup49 (1–220), Nup57 (1–220), Nup145N (1–220), Nup1 (1–720), and Nup60 (1–120), each being a continuous N-terminal fragment of the corresponding full-length protein. One complete set of 200 FG-nup fragments was generated, providing the initial conditions for our sole Kim2018 coarse-grained simulation. The C-terminal end of each FG-nup fragment was harmonically restrained to the anchor points prescribed by the Kim2018 structure[55]. Similar to our Lin2016 model, we used a purely steric, repulsive potential to represent the nuclear envelope of the yeast NPC and twenty grid-based potentials to represent its structured scaffold. The scaffold coordinates were taken from the integrative modeling database, entry PDBDEV ID 00000012 (associated with ref. 55).

Another version of a yeast NPC was designed separately, which we refer to as "Kim2018+". We employed the same nuclear envelope and scaffold potentials as was done for our Kim2018 model. The nup species, and their domains, for Kim2018+ were as follows: Nsp1 (residues 1–620), Nup159 (382–1141), Nup116 (1–760), Nup100 (1–560), Nup49 (1–220), Nup57 (1–220), Nup145N (1–220), all anchored at their N-terminal ends; Nup1 (1076–357), and Nup60 (539–420), which were anchored at their C-terminal ends; previously unconsidered nup species Nup42 (eight copies, residues 1–382), Nup2 (sixteen copies, residues 1–720); and linker 'connector' domains Nup116 (561–815), Nup100 (761–965), Nup145N (221–458), whose N- and C-terminal ends were restrained but were otherwise fully flexible. Supplementary Table 3 compares the FG-nup species' copy-numbers, lengths, and amino-acid ranges present for NPC models in our paper (Kim2018, Kim2018+) and one from Huan et al.[35], which we refer to as "Huan2020."

We note that our Kim2018 yeast NPC model does not include FG-nups Nup42 or Nup2, which are present in the Huang2020 model. Furthermore, FG-nups Nup1 and Nup60 are anchored from the C-terminal end in Kim2018 but from their N-terminal end in Huang2020. Notably, Huang2020 contains sixteen copies of Nup1, whereas our Kim2018 model only has eight. The starting point of Nup159 is different for Kim2018 (residue 1) versus Huang2020 (residue 388). Lastly, the domains of Nup100/Nup116/Nup145N are quite a bit shorter for Kim2018 compared to Huang2020. Combined, these differences mean the total mass of the FG-nup domains of the Huang2020 yeast NPC model is about 30% greater than the total mass of the domains present in our original Kim2018 model.

The updated yeast NPC model, which we refer to as Kim2018+, is much more closely related to Huang2020 than our original Kim2018 model. FG-nups Nup42 and Nup2 are included in our Kim2018+ model, with the same stoichiometry and domain lengths as in Huang2020. Furthermore, linker domains of Nup100/Nup116/Nup145N are present in our Kim2018+ model, increasing those nups' lengths to similar lengths as present in Huang2020. Nup1 and Nup60 are tethered from their N-terminal ends for Kim2018+, as is done in Huang2020. And the starting points of Nup159 are close for Kim2018+ (residue 382) and Huang2020 (residue 388). Overall, the total mass of the FG-nup domains present in our updated Kim2018+ model is quite close to the total mass of Huang2020.

## CG simulations of the NPC models
All CG MD simulations were performed using the Atomic Resolution Brownian Dynamics (ARBD) package[47]. The grid-based potentials representing the nuclear envelope and the protein scaffold were fixed in space. Following ref. 49, the motions of CG beads representing disordered FG-nup mesh were described using a Langevin dynamics integrator with a timestep of 0.02 ps and the Langevin friction coefficient of $50\,ps^{-1}$. The above damping coefficient corresponds to a diffusion coefficient roughly 500 times greater than experimental measurements of individual amino acids[63]. Hence, we scale the timestep by 500 to obtain an effective timestep of 10 ps. Reporting dynamics using the

effective timestep is sensible for Rouse-like polymers where hydrodynamic interactions can be neglected. We chose to report our simulation results using the effective timestep because we found the average radius of gyration of the FG-nups in our Lin2016 model to scale with the FG-nup length following a scaling law associated with extended intrinsically disordered peptides (Supplementary Fig. 17), which exhibit low internal friction[64]. Furthermore, hydrodynamic interactions are expected to be negligible due to screening in a dense mesh[65].

The two replica Lin2016 systems were simulated for 7500 μs each. One replica of the Kim2018 system was simulated for 7500 μs. All three systems were run as described below. During the first 1000 μs, the temperature of the system was reduced from 600 to 298.15 K in 100 evenly-spaced steps. The temperature was kept at 298.15 K thereafter using a Langevin thermostat. The Kim2018+ system was simulated for 1500 μs, the first 100 μs of which the temperature was reduced from 600 to 298.15 K in ten evenly-spaced steps. The systems were simulated under periodic boundary conditions with a unit cell measuring $120 \times 120 \times 360\,nm^3$. Bead–bead and bead–potential forces were calculated every simulation step with a cutoff of 5.0 nm and a pairlist distance of 6.0 nm; the pairlist was updated every 500 steps. The potentials were linearly interpolated to calculate the forces they exerted on beads. The system's coordinates were recorded every 10,000 steps and used for further analysis.

## CG model of proteins used in passive transport simulations
Rigid-body models of the 13 globular proteins were generated by converting their all-atom crystallographic structures (PDB IDs 2HIU, 4PTI, 2SPZ, 1UBQ, 1F6M, 1F6S, 1EMA, 2ABH, 1ANF, 6ENL, 4HHB, 1G5Y, and 1U8F) to a one-bead-per-residue representation. Thus, each protein rigid body was a collection of CG beads fixed in relative position to each other. Just like a bead in an FG-nup chain, each bead of the rigid body interacted with the beads of the FG-nup mesh or with the potentials representing the protein scaffold or the lipid membrane. At each timestep, the forces (and corresponding torques) on the constituent rigid-body beads were added up to obtain the net force (and torque) on the rigid body.

The translational and rotational diffusion coefficients of each protein rigid body were obtained from the atomic coordinates of the protein using the HYDROPRO software package[66]. Only diagonal elements of the diffusion tensor returned by HYDROPRO were used when integrating the equations of motion.

To validate our rigid-body approximation, we simulated two proteins—ubiquitin and GFP—using the all-atom molecular dynamics method, as described in Supplementary Methods. Over the course of a 400 ns all-atom simulation, each protein's root-mean-squared deviation with respect to the crystal structure coordinates converged to a value below 3 Å (Supplementary Fig. 18). Furthermore, their principal semi-axes fluctuated on the order of ±1 Å (Supplementary Fig. 18), which is well below the resolution of our rigid-body approximation, which justifies its use for the description of folded proteins.

## CG simulations of proteins diffusion through the NPC
In our simulations of single rigid-body protein diffusion through the Lin2016 structure, the motion of FG-nup was modeled using Langevin dynamics with an effective timestep of 10 ps, consistent with the original formulation of the CG model[48,49]. The protein diffusion coefficients along and about the protein principal axes were multiplied by a factor of 500 to make the protein diffusion timescale consistent with that of individual amino acids of the FG-nup mesh. The enhanced diffusion coefficient precluded the use of a Langevin dynamics integration scheme for the proteins, which would generate unreasonable persistent path lengths up to 6.2 nm for the largest protein—a significant fraction of the length of the central channel of the NPC. Hence, we update the coordinates of the protein rigid body each timestep

using a Brownian dynamics algorithm with an effective timestep of 10 ps so that the direction of the protein motion is uncorrelated between steps.

In addition to the potential representing the NPC scaffold and the membrane envelope, a rigid-body protein was also subject to a confinement potential that was zero within a cylinder centered at the origin. Outside that cylinder, a harmonic potential acted on the protein's center-of-mass with a spring constant of 0.01 kcal mol$^{-1}$Å$^{-2}$. The simulations of protein diffusion were performed using confinement potentials of 50 and 25 nm radius, the height of the cylindrical potential was 120 nm in both cases. The forces between the beads of the rigid bodies were calculated every timestep with a cutoff of 5 nm and a pairlist distance of 9 nm. The forces exerted on the rigid bodies by the lipid, scaffold, and constraint potentials were calculated by linearly interpolating the grid-based potentials at every step.

For each of the 13 globular protein species represented by rigid bodies, two replica systems were simulated differing only by the initial position of the protein, which was positioned the same (50 nm) distance away from the midplane along the pore axis in either direction. Each system was simulated for 6000 μs (12,000 μs per species). The coordinates of the NPC beads and of the protein rigid body were recorded every 2500 steps (25 ns). Additional simulations of protein diffusion were performed in the presence of the nuclear envelope and protein scaffold potentials only, i.e., in a model system containing no disordered FG-nup mesh.

Similar methods were used to simulate the simultaneous diffusion of two proteins—ubiquitin and GFP—through the Lin2016 structure. The simulations were carried out under the 50 nm-radius confinement potential. Each protein interacted with all components of the system, including the other protein. Two replica simulations were performed, lasting 2300 μs each. In the first replica system, the two proteins were initially located on opposite sides of the NPC, 50 nm away from the midplane, along the pore axis. The initial locations of the proteins were swapped in the second replica simulation. The coordinates of the proteins and of the FG-nup mesh were recorded every 25 ns.

## Density map analysis

The nucleoporin density maps were generated using the final 6500 μs of both CG simulations, taking into account the eight-fold rotational symmetry of the NPC about the pore axis, and the two-fold reflection symmetry with respect to the midplane of the nuclear envelope membrane. Each FG-nup bead was assigned a radius of 0.3 nm and a mass of 120 Da. The average 3D density of the FG-nups was calculated using `volmap` plugin of VMD, over a 120 × 120 × 360 nm$^3$ volume. The density maps were generated separately for each of the five nup species; the total nup density was calculated by adding together all the density maps for all species.

## Void analysis

To identify the volume that can accommodate, without a steric clash, a protein of radius $R_p$ for a given instantaneous configuration of the NPC, we first partitioned the whole system into cubic cells, each cell having a linear dimension equal to 6 Å for our PMF calculations and 1 Å for the connectivity analysis. Each cubic cell voxel, denoted as $(n_x, n_y, n_z)$, was assigned cartesian coordinates located at the voxel's center.

For a given configuration of the FG-nup mesh, we first determined the distance $R_{max}$ from the center of each voxel to the nearest nup particle surface (taking the nup particle's 3 Å radius into account) or to the nearest void analysis voxel of the non-zero scaffold, lipid or constraint potential (as defined in our CG MD simulation of protein transport), whichever was the closest. Each voxel was then classified as accessible for a protein type with radius $R_p$ if $R_{max} \geq R_p$; otherwise, it was classified as inaccessible to the protein. We implemented our algorithm using a cell-linked-lists data structure to store the positions

of FG-nups particles, achieving $O(N)$ computational complexity, where $N$ is the total number of voxels.

## Connectivity analysis

Results from the aforementioned void analysis were applied to perform connectivity analysis, in which we query the existence of a continuous path through the FG-nup mesh for each configuration. A layer of voxels above (at $z = +20$ nm) and below (at $z = -20$ nm) the FG-nup mesh were defined as the source and sink regions, respectively, for the continuous path search. Voxels between the source and the sink were identified as available (or not) according to the void analysis procedure. After identifying every void voxel for a given protein size, we used a union-find with a path compression algorithm to partition all accessible voxels into several disjoint sets, where any two neighboring accessible voxels were assigned to the same set. An open path crossing a given configuration of the FG-nup mesh was determined to exist in any of the above sets containing both source and sink voxels.

## Fokker-Planck formulation

The diffusion of a Brownian particle in potential $V(\vec{r})$ is described within the framework of the Smoluchowski equation

$$\frac{\partial \rho(\vec{r},t)}{\partial t} = \nabla \cdot \left[ D(\vec{r}) e^{-\beta V(\vec{r})} \nabla (e^{\beta V(\vec{r})} \rho(\vec{r},t)) \right], \qquad (3)$$

where $\rho(\vec{r},t)$ is the Brownian particle density at position $\vec{r}$ and time $t$, and $D(\vec{r})$ is the position-dependent diffusion coefficient. The geometry of the NPC allows the Smoluchowski equation to be approximated by a one-dimensional differential equation, where the position variable, $z$, is defined along the pore axis.

To solve the equation numerically, we discretize the coordinate space using $N + 2$ points, each separated by distance $d = 0.5$ Å from its neighbor. Each point has a coordinate $z_n = z_0 + nd$ with the two boundary points being at $z_0 = -600$ Å and $z_{N+1} = 200$ Å. Following the approach described in refs. 67 and 68, we rewrite Eq. (3) as

$$\frac{\partial \rho(z_n,t)}{\partial t} = k_{n+1 \to n} \rho(z_{n+1}, t) + k_{n-1 \to n} \rho(z_{n-1}, t)$$
$$- [k_{n \to n+1} + k_{n \to n-1}] \rho(z_n, t), \qquad (4)$$

where $k_{n \to m}$ is the transition rate from $z_n$ to $z_m$:

$$k_{n \to m} = \frac{[D(z_n) + D(z_m)]}{2d^2} \exp \left\{ -\beta \frac{[V(z_m) - V(z_n)]}{2} \right\}. \qquad (5)$$

Given the density at time $t_i$, we solved for $\rho$ a short time, $\Delta t = 10$ fs, later using the Euler method,

$$\rho(z_n, t_{i+1}) = \rho(z_n, t_i)$$
$$+ \left\{ k_{n+1 \to n} \rho(z_{n+1}, t_i) + k_{n-1 \to n} \rho(z_{n-1}, t_i) - [k_{n \to n+1} + k_{n \to n-1}] \rho(z_n, t_i) \right\} \Delta t. \qquad (6)$$

Our choice for $\Delta t = 10$ fs satisfies $D\Delta t/d^2 < 1$ even for the smallest (most diffusive) protein, ensuring convergence of the Euler method. To solve the first-passage time problem, we set an absorption boundary condition at $z_{N+1}$ and a reflecting boundary condition at $z_0$:

$$k_{0 \to 1} = k_{1 \to 0} = 0 \qquad (7)$$

$$\rho(z_{N+1},t) = 0 \qquad (8)$$

and the initial condition $\rho(z_n,t_0) = \delta(z_n - z')$ assuming the particle starts at $z'$. In our calculation, $z'$ is set to $-200$ Å. The first-passage time

(FPT) distribution $g(t)$ is

$$g(t) = \frac{d}{dt} \int_{z_0}^{z_{N+1}} dz\, \rho(z,t) \tag{9}$$

and the mean first-passage time (MFPT, denoted $\tau$) is then

$$\tau = \int_0^\infty dt\, g(t)t. \tag{10}$$

We approximate Eq. (9) and (10) using a finite difference/trapezoidal rule integration scheme.

## Calculation of PMF from void analysis

To estimate the MFPT from the Smoluchowski equation, the PMF $V(\vec{r})$ needs to be known a priori. Running MD simulations to estimate the PMF for very large proteins is not practical because the slow crossing time would limit sampling. Here we derived an algorithm to estimate the PMF without simulating actual protein translocation.

Neglecting rotational degrees of freedom, the partition function of a system containing FG-nups and one translocating protein can be written,

$$Z_{tot} = \int d\vec{Q}\, d\vec{r}\, e^{-\beta U(\vec{Q},\vec{r})} \equiv \int dz\, e^{-\beta F_{eff}(z)} \tag{11}$$

where $\vec{Q} = (\vec{Q}_0, \vec{Q}_1, \ldots, \vec{Q}_{N-1})$ is the vector of the total coordinates for $N$ FG-nup particles, $\vec{r} = (x,y,z)$ is the coordinate of the protein, $U(\vec{Q}, \vec{r})$ is the total potential energy of the system, and $F_{eff}(z)$ is the effective one-dimensional PMF that depends only on the $z$ coordinate of the protein. We rewrite $U(\vec{Q}, \vec{r}) = U_0(\vec{Q}) + U_{int}(\vec{Q}, \vec{r})$, where $U_0(\vec{Q})$ represents the interaction between FG-nup, and $U_{int}(\vec{Q}, \vec{r})$ represents the interaction between the protein and the FG-nups. Hence, we rewrite Eq. (11) as

$$Z_{tot} = \int dz \int d\vec{Q} \int dxdy\, e^{-\beta U_0(\vec{Q})} e^{-\beta U_{int}(\vec{Q},\vec{r})}$$
$$= \int dz \mathcal{Z}_0 \left\langle \int dxdy\, e^{-\beta U_{int}(\vec{Q},\vec{r})} \right\rangle_0, \tag{12}$$

where $\langle \ldots \rangle_0$ denotes the ensemble average over the configurations where no protein was present and $\mathcal{Z}_0 = \int d\vec{Q}\, e^{-\beta U_0}$ is a normalization constant. Combining Eqs. (11) and (12), we can write

$$F_{eff}(z) = -k_B T \ln \left\langle \int dxdy\, e^{-\beta U_{int}(\vec{Q},\vec{r})} \right\rangle_0 + F_0, \tag{13}$$

where $F_0$ comes from the normalization constant. Based on Eq. (13), to obtain $F_{eff}(z)$, one can simply generate a canonical ensemble for the system containing only FG-nup particles, and then evaluate the interaction energy for each configuration of the ensemble. The generation of the FG-nup ensemble can be done using brute-force Monte Carlo or MD simulations at a constant temperature, which is much more efficient than simulating proteins translocating through the NPC. Our derivation was under the assumption that there is only one copy of the protein; however, the resulting expression is valid for any dilute system where protein–protein interactions are negligible.

We further simplify Eq. (12) by approximating $U_{int}(\vec{Q}, \vec{r})$ with only steric interaction,

$$U_{int}(\vec{Q}, \vec{r}) = \begin{cases} \infty, & |\vec{r} - \vec{Q}_i| < q_i + R_p \\ 0, & \text{otherwise} \end{cases} \tag{14}$$

where $q_i$ is the radius of the FG-nup particle $i$, and $R_p$ is the radius of a spherical probe used for the void analysis (Supplementary Table 2).

Using Eq. (14), we express the partition function, $Z_{tot}$, and the effective PMF, $F_{eff}(z)$, in terms of the cross-section area $\mathcal{A}(z)$ available for a protein of radius $R_p$ at pore axis coordinate $z$:

$$Z_{tot} = \int dz\, \mathcal{Z}_0 \langle \mathcal{A}(z) \rangle_0 \tag{15}$$

and

$$F_{eff}(z) = -k_B T \ln \langle \mathcal{A}(z) \rangle_0 + F_0. \tag{16}$$

To efficiently evaluate Eq. (16) for proteins of arbitrary size, we first obtained an ensemble of protein-free FG-nup conformations by sampling instantaneous configurations from the 14 ms CG equilibration of the NPC system every microsecond. For each conformation, the FG-nup coordinates were binned into cells of a 3D grid with side-length $l_c = 6$ Å. For a given protein size and a nup conformation, we then evaluated the histogram $h(z_i)$ for the number of cells available to the protein at height $z_i$ using the void analysis algorithm described above. The histogram was then averaged over the ensemble of FG-nups, yielding the effective PMF

$$F_{eff}(z_i) = -k_B T \ln \langle h(z_i) \rangle_0 + k_B T \ln \left( \frac{\mathcal{A}_c}{l_c^2} \right). \tag{17}$$

where $\mathcal{A}_c$ is the cross-section of the cylindrical restraint potential used in our CG simulations. In the above expression, the second term ensures the PMF is zero far away from the pore. We separately estimated the PMF from the first and second replica trajectories of the FG-nups and found a negligible difference between them. The maxima of the PMFs differed by only 0.6% for a protein of radius 52 Å, the largest radius considered in this study, suggesting all PMF calculations have converged.

## Local diffusion approximations

Our expression for the PMF, Eq. (16), bears resemblance to the Fick-Jacobs equation describing the diffusion of a particle through an entropic barrier[69]. Following the heuristic argument of ref. [70], the reduction in the configuration space not only produces the entropic barrier but also has an effect on the diffusion coefficient. Hence, we approximate the diffusion coefficient $D(z)$ at cross-section $\mathcal{A}(z)$ to be

$$D(z) = \frac{D_0}{\left[ 1 + (dw(z)/dz)^2 \right]^{0.5}} \tag{18}$$

where the effective half width of the channel, $w(z)$, is defined as

$$\mathcal{A}(z) = h(z) \times l_c^2 = w(z)^2 \pi. \tag{19}$$

Using the expression for the effective PMF, Eq. (16), and for the position-dependent diffusion coefficient, Eq. (18), we numerically solved Eq. (3) for $\rho(z,t)$, $g(t)$, and MFPT $\tau$ using a custom C code.

The results of the Fokker-Planck MFPT analysis displayed in Figs. 5 and 6 were obtained using the Reguera and Rubi[70] model of the local diffusion coefficient, Eq. (18) above. To check if our results are sensitive to this particular choice of the local diffusion model, we have repeated our Fokker-Planck MFPT calculations using the local diffusion model of Zwanzig[69]:

$$D(z) = \frac{D_0}{\left[ 1 + 0.5(dw(z)/dz)^2 \right]}, \tag{20}$$

where $D_0$ and $w(z)$ have the same meaning as above. We found the choice of the local diffusion model to have little influence on the results of our MFPT analysis (Supplementary Fig. 19).

### Reporting summary
Further information on research design is available in the Nature Research Reporting Summary linked to this article.

## Data availability
The files needed to setup an annealing NPC simulation and a single-protein translocation simulation as were performed in this study have been deposited in the UIUC data bank under accession code IDB-3813848 (https://databank.illinois.edu/datasets/IDB-3813848). The simulation trajectory files and scaffold structures generated in this study have been deposited in the UIUC data bank under accession code IDB-5581194 (https://databank.illinois.edu/datasets/IDB-5581194). The data presented in Figs. 1e, 2e–h, 3e, 4b–d, f, g, i, j, 5a–d, f, h–j, and 6c–f are provided as a Source Data file ("NatComm_extraData.xlsx"). The analysis scripts are available upon reasonable request. And the following structures were used from the Protein Data Bank: PDB IDs 2HIU, 4PTI, 2SPZ, 1UBQ, 1F6M, 1F6S, 1EMA, 2ABH, 1ANF, 6ENL, 4HHB, 1G5Y, 1U8F.

## Code availability
The source code and the application examples for the void analysis method and numerical solution of the 1D Fokker-Planck equation are available at https://gitlab.engr.illinois.edu/tbgl/pubdata/void_analysis and https://gitlab.engr.illinois.edu/tbgl/pubdata/fokker_planck_1d, respectively.

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

## Acknowledgements

This work was supported by the Center for the Physics of Living Cells through the National Science Foundation grant PHY-1430124 to A.A. and by the National Institutes of Health through grants R01-GM137015 and P41-GM104601 to A.A. The supercomputer time was provided by the Extreme Science and Engineering Discovery Environment (allocation MCA05S028), the Blue Waters petascale supercomputer system (UIUC), and Leadership Resource Allocation MCB20012 on Frontera at the Texas Advanced Computing Center. Frontera is made possible by National Science Foundation award OAC-1818253. We thank Drs. Patrick Onck and Ali Ghavami for sharing files that described potentials of their coarse-grained model. A.A. would like to thank Dr. Cees Dekker for the invitation to join the nuclear transport field.

## Author contributions

D.W. and H.-Y.C. contributed equally as co-first authors of the study. D.W. and A.A. conceived the study. D.W. and C.M. built the experimentally derived NPC model. D.W. performed coarse-grained simulations. H.Y.-C. developed and performed void analysis and Fokker-Planck calculations. C.M. and A.A. assisted with analysis done by D.W. and H.-Y.C. All authors contributed to writing and editing the manuscript.

## Competing interests

The authors declare no competing interests.
