## [Peer Review File · Nature Communications]

Percolation transition prescribes protein size-specific barrier to passive transport through the nuclear pore complex.Reviewers' Comments:

Reviewer #1:

Remarks to the Author:

In this manuscript the authors give the results of coarse grained simulations and Fokker Planck modelling of the nuclear pore complex. Their approach is based on simulations of the FG-nups presents in the nuclear pore. Their objective is to understand the dynamics of this network and the ability of the nuclear pore to achieve passive transport of proteins. The main point tackled by this research is to test with quantitative simulations the existence of a size threshold for passive transport of nano-objects.

Their results show that the FG-nups network is highly dynamic. They observe that most of the contacts between FG-nups residues are not due to hydrophobic-hydrophobic interactions as previously proposed.

Using void analysis and Fokker-Planck modelling they demonstrate that the efficient diffusion of proteins through the FG-nups network can be understood by the presence of a continuous and fluctuating nanochannel through the mesh. As a consequence they show that there is no well defined size cutoff for diffusion of proteins.

The approach of this work is promising and the results presented in this manuscript are very informative for theoreticians but also for experimentalists interested in nucleocytoplasmic transport. I think this work is a crucial step forward for the research on nuclear pore complex and should be published with some modifications in Nature Communications.

Most of the following requests concern some important elements that should be added to the discussion and the results in order to show a better comparison with existing literature and increase the impact of this work.

1. The dynamics of the luminal ring /scaffold is not included in the simulations. Orders of magnitude or simulations of the NPC scaffold should be included in the discussion to show that the effects observed in this work (percolation transition) will not be impacted by fluctuations of the ring.
2. A more detailed comparison with the (large) literature on the FG-nups dynamics and simulations should be included to explain the differences with previous numerical results. In particular the importance of the hydrophobicity of the residue in the FG-FG interactions and domain/liquid-phase formation.
3. The dynamics of the transported proteins are not included in the simulation. Orders of magnitude should be included in the discussion to explain the reason for this choice. More over, do they observe the same crossing timescale in presence of two proteins instead of one ?
4. The authors claim that in presence of the FG-nups they observe a crossover in the range 30 to 90kDa between a power law and an exponential regime. The power law is well demonstrated in Fig 2 and 5 but the exponential regime is unclear on Fig. 5d and S11d. From what we see on the Figures, it could also be another power law. This crossover has important connections to understand the dynamics of the transport and the existence of a size cutoff for transport. Therefore the fits of the exponential regime should appear on the Figures. The discussion on p17 about the crossover should be expanded and the parameters extracted from the fitting procedure should be analysed and discussed in more details.
5. Finally, the title of the manuscript implies that there is a size limit for transport whereas the core of the article is that this size limit is not a well-defined criterion since the transit time follows a power law and is induced by the fluctuations of the FG-nups network. I think the title should be changed as it is misleading about the content of the manuscript

Typos :

p13 : "exists" -> "exits"

p13 : "μ" -> "μs"

p13 : "that that" -> "that"

p14 : "the the" -> "the"

p18 Figure 5f : "panel f" -> "panel d"

p19 : multiple errors of panel label about Fig5

Reviewer #2:

Remarks to the Author:

Peer review NCOMMS-21-49443: Percolation transition determines protein size limit for passive transport through the nuclear pore complex

In this work, the authors describe a novel method to characterize passive transport through a Nuclear Pore Complex (NPC). The authors start by describing the coarse-grained NPC model that is used throughout the manuscript. The authors report on the equilibrium structure of the NPC complex, with the key observation being that the FG-Nups within the NPC central channel form a connected meshwork, held together by non-specific interactions between residues.

By tracking single inert particles through the NPC model, the authors investigate the scaling of the translocation barrier for a large range of particle sizes. Using Boltzmann inversion, the authors show that the translocation barrier increases with protein mass, while there is only a modest effect of protein size when there is no FG-Nup meshwork present. Next to that, the authors show that the mean first passage time (MFPT) scales with the protein volume (m^3), while it scales with the protein radius ($m^{1/3}$) in the absence of FG-Nups in the NPC central channel.

The authors then continue to investigate NPC barrier crossing times and relate the protein crossing times to the free diffusion coefficients of the translocating particles. The authors highlight that the process of passive transport through the NPC consists of many unsuccessful translocation attempts interrupted by infrequent successful translocations.

Based on this result, and because of the observation that passive diffusion of large particles is a rare event, the authors then describe an alternative method using a void analysis on the equilibration trajectories. This method reproduces PMF curves which are in good quantitative agreement with the PMF curves from the brute force simulations. The authors note that the void analysis has better statistical sampling than the MD simulations as it is not depending on the specific translocation path of the diffusing protein.

The authors then continue to use the PMF curves obtained with the void analysis to estimate transport times by numerically solving the Fokker-Planck equation, again in agreement with CG simulation data. They suggest the presence of a transition from a soft to hard translocation barrier with increasing protein mass.

The paper is well-written and displays an innovative combination of void analysis and the Fokker-Planck theory to describe (passive) nuclear transport and could therefore potentially be of high interest. However, the choice of NPC scaffold and corresponding FG-Nups leads to an underprediction of the actual FG-Nup concentration, which affects the reported readouts and limits the impact of the results for the wider NPC community. The authors should properly remedy this before the paper can be accepted for publication in Nature Communications.

Major concerns:

The authors refer to the used NPC model as a 'complete' NPC structure. However, the used structure (from Lin/Hoelz) is a 'composite' NPC consisting of *C. thermophilum* Nucleoporin complexes fit into a cryo-EM map of the human NPC. The authors here use a set of five intrinsically disordered FG-Nups from the NPC central channel of *C. thermophilum*, omitting several cytoplasmic and nucleoplasmic FG-Nups. As a result, the real FG-Nup mass concentration is underestimated and the spatial distribution of the FG-Nups inside the NPC is considerably affected. This will strongly influence the reported

readouts (PMF, transport times, etc.) as these are all based on the concentration and organization of the FG-Nups inside the channel.

For example, the reported densities in the NPC central channel are quite different from earlier reported values (e.g. Ghavami [1], Huang [2], Peyro [3]). These differences should be discussed in light of the different stoichiometries of the composite NPC (*C. thermophilium* versus human/yeast): there are 48 copies of Nsp1 and Nup62 in yeast for instance, not 32, and also Nup98 (human) and Nup100+116+145N (yeast) have 48 copies rather than 32. Furthermore, no cytoplasmic and nucleoplasmic FG-Nups are account for, which are known to play an important role in establishing the permeability barrier.

Nup145N is a homolog of human Nup98 and yeast Nup100/116/145N, so one would expect a similar behavior. However, Nup145N seems to have little interactions with the other central channel FG-Nups (Fig. 1c and 5g) in contrast to yeast for instance where Nup100/116/145N collectively form a dense ring-shaped region, which plays an important role in controlling passive diffusion. The authors should address these discrepancies. For instance, what is the average charge over hydrophobicity ratio (C/H) of the FG-Nups in the current study and how does this compare with the homolog-structures in yeast and human NPCs? In addition, Nup145N is anchored far towards the cytoplasmic and nucleoplasmic side (Fig. 5g) and not centrally anchored like the human Nup98 and yeast Nup100/116/145N homologs. All these aspects should be discussed as well as these likely play an important role in the distribution of the FG-Nups and thus the readouts of the void model and the Fokker-Planck analysis.

The authors find that most of the interchain contacts were formed between a hydrophobic and a polar residue (45%), about twice as likely as a hydrophobic-hydrophobic or polar-polar contact. This is in contrast to what is reported in many other studies, both experimental and theoretical, and again raises the question whether the stoichiometry of the composite NPC structure used is biologically relevant.

In recent years it has become undeniably evident that, at all times, a significant concentration of nuclear transport receptors is present in the NPC (see e.g. work from the groups of Lim and Rout). Considering that a large part of the paper depends on PMF curves obtained from available voids, this potentially has a drastic effect on the observed transport times. This is ignored in the current paper.

The current study predicts a more-or-less linear dependence of the transport times (MFPT) with the mass of the cargo proteins. Experimental data from the groups of Veenhoff and Rout (refs. 6 and 8 from the paper, resp.), however, show a cubic dependence of the transport times on the molecular mass. How can these seemingly contrasting observations be justified?

Minor concerns:

Some additional concerns are listed below:

- The authors report simulation times using an estimated effective timestep. The used time scaling is justified by the claim that the FG-Nups are fully extended IDPs. However, the FG-Nups (especially in the NPC central channel) are not generally considered as fully extended IDPs (see the work of Yamada et al. 2010), and thus the used timestep rescaling might not be appropriate.
- More details should be given on the implementation of the coarse-grained model. The authors write: ‘... the beads were connected into a polymer chain via a harmonic spring potential, see Refs. 49 and 50 for details’. From this it is not clear whether the angular and dihedral potentials (originally present in the CG model) are included or not.
- The authors have verified the implementation of the coarse-grained model from a brush simulation. However, it is not clear how the height is determined, since this is not trivial for brushes. It would also be good to do a check of the Stokes’ radii of several yeast FG-Nups to assess whether single-chain properties are correctly retrieved (see the work of Yamada et al. 2010).

[1] Ali Ghavami, Liesbeth M. Veenhoff, Erik van der Giessen, Patrick R. Onck, Probing the Disordered Domain of the Nuclear Pore Complex through Coarse-Grained Molecular Dynamics Simulations. *Biophysical Journal* 107, 2014, p. 1393-1402. DOI: 10.1016/j.bpj.2014.07.060

[2] Kai Huang, Mario Tagliacruzchi, Sung Hyun Park, Yitzhak Rabin, Igal Szleifer, Nanocompartmentalization of the Nuclear Pore Lumen. *Biophysical Journal* 118, 2019, p. 219-231. DOI: 10.1016/j.bpj.2019.11.024

[3] Mohaddeseh Peyro, Mohammad Soheilypour, Ali Ghavami, Mohammad R. K. Mofrad, Nucleoporin's Like Charge Regions Are Major Regulators of FG Coverage and Dynamics Inside the Nuclear Pore Complex. *PLoS ONE* 10(12): e0143745. doi:10.1371/journal.pone.0143745

Reviewer #3:

Remarks to the Author:

The authors perform extensive large-scale coarse-grained (CG) Molecular Dynamics (actually Brownian Dynamics) simulations on the routes and rate of transport of several protein molecules through Nuclear Pore Complex (NPC). The existence of a polymer mesh made of natively unstructured proteins (nucleoporins) that are grafted to the pore scaffold forces the translocating protein to find its way through holes in the mesh. This becomes more difficult to do as the protein size increases, for reasons elucidated in the manuscript, resulting in a "hard" (virtually impenetrable) barrier for proteins above a certain size. All of this is interesting enough from a mechanistic point of view, and nicely supported by an approximate theoretical model that gives results in surprisingly close agreement with the full CG simulations.

In general, unassisted ("passive") diffusion of relatively small proteins is not what makes the NPC interesting. In fact, large proteins and strands of mRNA can translocate through the NPC, but only if they attach themselves to special globular receptor proteins, sometimes called Nuclear Transport Factors (NTFs). These NTFs have hydrophobic surfaces that bind transiently to segments of the polymer mesh (i.e., nucleoporin) proteins. Without such binding events, the NPC does not allow for selective transport of large protein or RNA cargos, which is its critical physiological function. Such transient binding effects are not considered in the authors' model, as they acknowledge in the Conclusion section.

So, this paper does not resolve the fundamental mystery of how the NPC operates, namely, how it enables the transport of large cargo molecules in/out of the nucleus in a highly selective and highly efficient (rate of translocation on the order of 1 cargo molecule/ms) manner. Nevertheless, it presents a carefully crafted study of passive diffusion through the NPC, which provides novel insights into the mechanism of this process.

The following questions should be answered prior to publication:

1) Where does Importin- β (a representative member of the karyopherin [NTF] receptor family) fall on this protein size range considered in the present work (e.g., in Fig. 5d). Presumably, it is above the cutoff size for no passive transport.

2) Diffusion constant $D(z)$ profile: Explain Eq. 17, namely, the dependence on cross sectional area at point z on the pore axis. Where does this functional form come from? If there is no rigorous justification for this profile, how sensitive are the results obtained for Mean First Passage Times (MFPTs), for example, to the functional form employed here? This question is relevant since the authors claim quantitative success of this 1-d drift-diffusion model in explaining their many-atom CG BD simulation data.

3) Potential of Mean Force (PMF) calculation via "Boltzmann inversion" (p. 10): I presume the CG BD trajectory runs long enough for the protein molecule to cross through the pore multiple times, and spend significant amounts of time in the reservoirs on either side of the pore. Please confirm.

4) Usage of the cylindrical confining potential. This is done to accelerate the calculation of MFPTs and transport rate constants by reducing the amount of the time the protein bounces around in the reservoirs. Fine, but, particularly for calculation of experimentally observable properties like the transport rate, this confinement potential should not affect the predicted results. Has this been confirmed, presumably by increasing the value of the confinement radius R_{con} until the extracted rate constant does not change with further increase of R_{con} ?

RESPONSE TO REVIEWER COMMENTS

Reviewer #1 (Remarks to the Author):

In this manuscript the authors gives the results of coarse grained simulations and Fokker Planck modelling of the nuclear pore complex. Their approach is based on simulations of the FG-nups presents in the nuclear pore. Their objective is to understand the dynamics of this network and the ability of the nuclear pore to achieve passive transport of proteins. The main point tackled by this research is to test with quantitative simulations the existence of a size threshold for passive transport of nano-objects.

Their results shows that the FG-nups networks is highly dynamics. They observe that most of the contacts between FG-nups residues are not due to hydrophobic-hydrophobic interactions as previously proposed. Using void analysis and Fokker-Planck modelling they demonstrate that the efficient diffusion of proteins through the FG-nups network can be understood by the presence of a continuous and fluctuating nanochannel through the mesh. As a consequence they show that there is no well defined size cutoff for diffusion of proteins.

The approach of this work is promising and the results presented in this manuscript are very informative for theoreticians but also for experimentalists interested in nucleocytoplasmic transport. I think this work is a crucial step forward for the research on nuclear pore complex and should be published with some modifications in Nature Communications.

Our response: We thank the reviewer for the enthusiastic review of our work and for the insightful comments that have helped us to place our study in the context of previous work.

Most of the following requests concerns some important elements that should be added to the discussion and the results in order to show a better comparison with existing literature and increase the impact of this work.

1. The dynamics of the luminal ring /scaffold is not included in the simulations. Orders of magnitude or simulations of the NPC scaffold should be included in the discussion to show that the effects observed in this work (percolation transition) will not be impacted by fluctuations of the ring.

Our response: According to the results of AFM studies, reviewed in [Sakiyama2017], the diameter of the central ring of the NPC scaffold can fluctuate by 10% on the seconds timescale. Assuming the stoichiometry of the assembly is preserved, such fluctuations would result in a ~20% change of the FG-nup mesh density. In response to the comments of reviewer 2, we have carried out CG simulations of another NPC structure (referred to as

Kim2018), which happen to have about a ~15% denser FG-nup mesh than the structure we have reported in the previous version of the manuscript (Lin2016). Our void analysis of the new, denser model has found the percolation transition to determine a scaling crossover from a soft to a hard barrier (just as for the older structure), but the actual numerical values of the threshold radius to shrink by several angstroms, please see Figure 6f of the revised manuscript. Thus, similar changes in the percolation threshold / protein selectivity can be expected for an NPC undergoing structural fluctuations. For NPC structures having a wider central channel, as seen in [Schuller2021, Zimmerli2021, Akey2022], we anticipate the percolation/scaling transition to shift toward larger protein sizes, but we expect such a transition to occur nonetheless.

Changes made to the manuscript:

Pages 23-24: “Thus, we have found a percolation transition to prescribe size limit for passive transport through two models of NPC built using two independent experimental structures. Although the precise value of the percolation transition is found to depend on the overall density of the FG-nup mesh, the physical mechanism enabling protein size selection remained unchanged and hence we believe it to be a general feature of a disordered mesh. For NPC structures featuring wider scaffolds but similar FG-nup stoichiometry, we expect the percolation threshold to shift toward larger protein sizes, compared to the values determined for the Lin2016 and Kim2018 structures. ... Similar shifts in the percolation transition can be expected for the NPC scaffolds that can change their size over time, as seen in the AFM studies [58]. As a reference, we note that a ~10% change in the scaffold ring diameter would amount to a ~20% change of the FG-nup density, a density change of the same magnitude as that between the Lin2016 and Kim2018 structures.”

2. A more detailed comparison with the (large) literature on the FG-nups dynamics and simulations should be included to explain the differences with previous numerical results. In particular the importance of the hydrophobicity of the residue in the FG-FG interactions and domain/liquid-phase formation.

Our response: Our response to this comment has three parts that are provided below

Response 2a. The revised version of our manuscript now includes a side-by-side quantitative comparison of the FG-nup densities seen in our work to the results of the previous two computational studies, see Supplementary Fig. 1 (reproduced below). In Supplementary Fig. 2, we now also provide the charge-to-hydrophobicity (C/H) ratio of FG-nup fragments in our Lin2016 and Kim2018 models, as well as the average density due to hydrophobic residues alone. We have also extended our comparison to previous numerical results and discussed the importance of hydrophobicity.

Supplementary Fig. 1: Density and stoichiometry of FG-nups across recent computational models of NPC.

Changes made to the manuscript:

Added new Supplementary Fig. 1 (see above)

Page 7: “In comparison to earlier computational studies of model NPC systems [49,50], Supplementary Fig. 1a, our local FG-nup density map has similar values in the middle of the central channel but shows much lower values near the anchor points, which we attribute to differences in both the scaffold dimensions and the nups’ stoichiometry. Our density map is comparable to that reported in a more recent computation study [35], Supplementary Fig. 1a.”

Page 8: “We note, however, that the FG-nup segments present in our Lin2016 model all have a relatively low charge-to-hydrophobicity ratio, Supplementary Fig. 2c.”

Pages 22-23: “Contact analysis of the Kim2018 equilibration trajectory shows that, just as with the Lin2016 model, a hydrophobic-polar pairing is more likely than a hydrophobic-hydrophobic or polar-polar pairing, Supplementary Fig. 2. We would like to point out that the contacts that involve hydrophobic residues make up over 60% of all contacts observed over the course of CG simulations of both models. And although we find a hydrophobic-polar

residue contact to be the most prevalent, the hydrophobic residues contribute, overall, the largest fraction of inter-nup contacts, in comparison to either charged or polar residues.”

Response 2b. Indeed, many previous numerical studies have concluded that hydrophobic residues are important, for example, by mutating hydrophobic residues to polar serine and observing a major change in the overall FG-nup density. In terms of contacts specifically, we would like to point out that, according to our Supplementary Fig. 2f (reproduced below), contacts involving hydrophobic residues (HH, HP, HC) make up over 60% of all contacts present. In our literature search, we did not find contact analysis done in the same pairwise manner presented above. While a hydrophobic-polar pair is the most likely for our Kim2018 CG simulations, hydrophobic residues contribute the most to inter-nup contacts overall compared to either charged or polar residues.

For reference, below we provide Figure 3 from [Ghavami2014], which graphically visualizes the densities due to all FG-nup residues (orange), charged residues (red) and hydrophobic residues (green).

Response Fig. 1: The three-dimensional density distribution of different amino acids inside the NPC (Figure 3 from [Ghavami2014]). Density of all FG-nup residues is shown in orange, charged residues in red and hydrophobic residues in green.

Our density maps exhibit a qualitatively similar property, in that the volume occupied by hydrophobic residues spans a similar space to all FG-nups residues at a lower overall density. From our results, we can compare Fig. 1b (all for Lin2016) to Supplementary Fig. 2d (hydrophobic for Lin2016), as well as Fig. 6b (all for Kim2018) to Supplementary Fig. 2h (hydrophobic for Kim2018). From our simulations, all residues reach an upper density limit of about 130 mg/mL whereas hydrophobic residues alone reach about 65 mg/mL, but span a similar overall volume.

Supplementary Fig. 2: Properties of the FG-nup mesh.

Changes made to the manuscript:

Included additional panels in Supplementary Fig. 2 (see above)

Page 7: “We find the spatial distribution of the hydrophobic residues of the FG-nup, Supplementary Fig. 2d, to follow the overall FG-nup density, similar to observations reported in Ref. 49.”

Response 2c. The phase separation of FG-nups has indeed been observed in experimental studies. Schmidt et al. in particular observed Nup98 to phase separate on its own at a density of about 250 mg/mL [Schmidt2015]. The FG-nups in our coarse-grained simulations reach an upper density limit of about 130 mg/mL, suggesting that they are not dense enough to undergo phase separation. Theoretical studies have also investigated FG-nup phase separation, such as the result from [Zilman2018], which determined the phase diagram of FG-nup using terms of FG-nup concentration and cohesiveness.

Changes made to the manuscript:

Page 7: “Experimentally, Nup98 was observed to undergo a phase separation at a density of about 250 mg/mL [52], while theoretical work [53] has examined the phase separation propensity as a function of cohesive interactions between FG-nups. In our simulations, the local density of the FG-nups remained under 130 mg/mL and we did not observe any signs of phase separation.”

3. *The dynamics of the transported proteins are not included in the simulation. Orders of magnitude should be included in the discussion to explain the reason for this choice. Moreover, do they observe the same crossing timescale in presence of two proteins instead of one ?*

Our response: Our response to this comment has two parts that are provided below

Response 3a. In response to this comment, two globular proteins – ubiquitin (1UBQ) and GFP (1EMA) – were simulated using explicit solvent all-atom molecular dynamics to observe how their structure changes in time. The results of these simulations are summarized in Supplementary Fig. 17 (reproduced below) whereas Supplementary Methods provide a complete description of the simulation protocols. According to our simulations, the fluctuation in the protein structure is well below the resolution of our coarse-grained model, which justifies our rigid-body approximation for the folded proteins.

Supplementary Fig. 17: Structural fluctuations of globular proteins in all-atom MD simulations.

Changes made to the manuscript:

Added new Supplementary Fig. 17 (reproduce above).

Added the following new section to the Supplementary Materials: “Supplementary Methods: All-atom MD simulations of ubiquitin and GFP.”

Pages 31: “To validate our rigid body approximation we simulated two proteins—ubiquitin and GFP—using the all-atom molecular dynamics method, as described in Supplementary Methods. Over the course of a 400 ns all-atom simulation, each protein’s root mean-squared deviation with respect to the crystal structure coordinates converged to a value below 3 Å, Supplementary Fig. 17. Furthermore, their principal semi-axes fluctuated on the order of ± 1 Å, Supplementary Fig. 17, which is well below the resolution of our rigid-body approximation, which justifies its use for the description of folded proteins.”

Response 3b. In response to the second part of the comment, we carried out additional simulations where two proteins, ubiquitin and GFP, could diffuse simultaneously through the NPC. Two replica simulations were performed (both containing two proteins) for 2,300 μs each, using the same simulation protocol as our single protein simulations. With both proteins present, we obtained the following MFPT values: $127.66 \pm 19.65 \mu\text{s}$ (ubiquitin), $352.71 \pm 181.01 \mu\text{s}$ (GFP). Whereas, under the same conditions with only one protein present: $142.49 \pm 12.16 \mu\text{s}$ (ubiquitin), $311.55 \pm 52.44 \mu\text{s}$ (GFP). We provide below a figure tracing the proteins’ z positions throughout the 2-protein simulations (new Supplementary Fig. 8):

Supplementary Fig. 8: Simultaneous passive diffusion of ubiquitin and GFP across the Lin2016 model of the NPC.

Changes made to the manuscript:

Added new Supplementary Fig. 8 (reproduce above).

Page 12: “Test simulations featuring two proteins (ubiquitin and GFP) diffusing simultaneously through the NPC, Supplementary Fig. 8, yielded MFPT values ($127.66 \pm 19.65 \mu\text{s}$, ubiquitin; $352.71 \pm 181.01 \mu\text{s}$, GFP) within one standard deviation of the values obtained from our single protein simulations ($142.49 \pm 12.16 \mu\text{s}$, ubiquitin; $311.55 \pm 52.44 \mu\text{s}$, GFP).”

4. The authors claim that in presence of the FG-nups they observe a crossover in the range 30 to 90kDa between a power law and an exponential regime. The power law is well demonstrated in Fig 2 and 5 but the exponential regime is unclear on Fig. 5d and S11d. From what we see on the Figures, it could also be another power law. This crossover has important connections to understand the dynamics of the transport and the existence of a size cutoff for transport. Therefore the fits of the exponential regime should appear on the Figures. The discussion on p17 about the crossover should be expanded and the parameters extracted from the fitting procedure should be analysed and discussed in more detail.

Our response: We would like to emphasize that a single functional form (Eq. 1 in the revised manuscript) was used to fit the whole range of MFPT data. The functional form is based on the transition frequency formula $\tau \sim 1/De^H$ where D is the effective diffusion coefficient and H is the effective barrier height between the initial and the final states. The diffusion coefficient is inversely proportional to the protein radius, and we assume that the effective barrier height scales as $H \sim (R_p + 3 \text{ \AA})^\alpha$, so we have the fitting functional form $\tau_0/De^{[(R_p+3 \text{ \AA})/R_0]^\alpha}$ where τ_0 is the proportional constant and R_0 in the characteristic length scale. When $R_p \ll R_0$, the exponential can be Taylor expanded using one or two terms, the power-law dominates. Otherwise, the exponential term takes over the relation.

To demonstrate that our fitting procedure works for the whole range of protein radii, we obtained additional data points in the limit of small and large probe radius values using our theoretical model. Plotting MFPT versus probe radius using either log-log and log-linear scales (see Response Fig. 2 below) shows that neither piecewise power law nor simple exponential law can fit the data well (the small radius fit should be a power law as the exponential relation doesn't pass through zero). Instead, our functional expression fit the MFPT dependence in the whole range of the probe radius values.

Response Fig. 2: Mean first-passage time versus probe radius, plotted on log-log (left column) and log-linear (right column) scales. Red lines connect data points, black lines show a fit obtained using Eq. 1 of the revised manuscript. R indicates the radius of the confinement potential.

Changes made to the manuscript:

We have expanded our description of the fitting procedures in the main text, please see Page 18 of the main text.

We have updated Figure panels 5d and S13d (and reproduced below as Response Fig. 3) to include additional Fokker-Planck data points for the two smallest and the two largest proteins, as well as a fit to all the Fokker-Planck results (black line).

Response Fig. 3: Updated mean first-passage time vs protein molecular mass, panels for Fig. 5d (left) and Supplementary Fig. 13d (right).

5. Finally, the title of the manuscript implies that there is a size limit for transport whereas the core of the article is that this size limit is not a well-defined criterion since the transit time follows a power law and is induced by the fluctuations of the FG-nups network. I think the title should be changed as it is misleading about the content of the manuscript.

Our response: We agree with the reviewer and changed the title of the manuscript to:

"Percolation transition **prescribes** protein size-**specific barrier to** passive transport through the nuclear pore complex"

6. Typos : p13 : "exists" -> "exits" p13 : "μ" -> "µ" p13 : "that that" -> "that" p14 : "the the" -> "the" p18 Figure 5f : "panel f" -> "panel d" p19 : multiple errors of panel label about Fig5

Our response: Thank you for pointing out these typos in the main text. We have fixed them.

References used in the response to Reviewer 1:

[Sakiyama2017] = Sakiyama, Yusuke, Radhakrishnan Panatala, and Roderick YH Lim. "Structural dynamics of the nuclear pore complex." In *Seminars in cell & developmental biology*, vol. 68, pp. 27-33. Academic Press, 2017.

[Schuller2021] = Schuller, Anthony P., Matthias Wojtynek, David Mankus, Meltem Tatli, Rafael Kronenberg-Tenga, Saroj G. Regmi, Phat V. Dip et al. "The cellular environment shapes the nuclear pore complex architecture." *Nature* 598, no. 7882 (2021): 667-671.

[Zimmerli2021] = Zimmerli, Christian E., Matteo Allegretti, Vasileios Rantos, Sara K. Goetz, Agnieszka Obarska-Kosinska, Ievgeniia Zagoriy, Aliaksandr Halavatyi et al. "Nuclear pores dilate and constrict in cellulose." *Science* 374, no. 6573 (2021): eabd9776.

[Akey2022] = Akey, Christopher W., Digvijay Singh, Christna Ouch, Ignacia Echeverria, Ilona Nudelman, Joseph M. Varberg, Zulin Yu et al. "Comprehensive structure and functional adaptations of the yeast nuclear pore complex." *Cell* (2022).

[Ghavami2014] = Ghavami, Ali, Liesbeth M. Veenhoff, Erik van der Giessen, and Patrick R. Onck. "Probing the disordered domain of the nuclear pore complex through coarse-grained molecular dynamics simulations." *Biophysical journal* 107, no. 6 (2014): 1393-1402.

[Schmidt2015] = Schmidt, Hermann Broder, and Dirk Görlich. "Nup98 FG domains from diverse species spontaneously phase-separate into particles with nuclear pore-like permselectivity." *Elife* 4 (2015): e04251.

[Zilman2018] = Zilman, Anton. "Aggregation, phase separation and spatial morphologies of the assemblies of FG nucleoporins." *Journal of molecular biology* 430, no. 23 (2018): 4730-4740.

Reviewer #2 (Remarks to the Author):

Peer review NCOMMS-21-49443: Percolation transition determines protein size limit for passive transport through the nuclear pore complex

In this work, the authors describe a novel method to characterize passive transport through a Nuclear Pore Complex (NPC). The authors start by describing the coarse-grained NPC model that is used throughout the manuscript. The authors report on the equilibrium structure of the NPC complex, with the key observation being that the FG-Nups within the NPC central channel form a connected meshwork, held together by non-specific interactions between residues.

By tracking single inert particles through the NPC model, the authors investigate the scaling of the translocation barrier for a large range of particle sizes. Using Boltzmann inversion, the authors show that the translocation barrier increases with protein mass, while there is only a modest effect of protein size when there is no FG-Nup meshwork present. Next to that, the authors show that the mean first passage time (MFPT) scales with the protein volume (m^1), while it scales with the protein radius ($m^{1/3}$) in the absence of FG-Nups in the NPC central channel.

The authors then continue to investigate NPC barrier crossing times and relate the protein crossing times to the free diffusion coefficients of the translocating particles. The authors highlight that the process of passive transport through the NPC consists of many unsuccessful translocation attempts interrupted by infrequent successful translocations.

Based on this result, and because of the observation that passive diffusion of large particles is a rare event, the authors then describe an alternative method using a void analysis on the equilibration trajectories. This method reproduces PMF curves which are in good quantitative agreement with the PMF curves from the brute force simulations. The authors note that the void analysis has better statistical sampling than the MD simulations as it is not depending on the specific translocation path of the diffusing protein.

The authors then continue to use the PMF curves obtained with the void analysis to estimate transport times by numerically solving the Fokker-Planck equation, again in agreement with CG simulation data. They suggest the presence of a transition from a soft to hard translocation barrier with increasing protein mass.

The paper is well-written and displays an innovative combination of void analysis and the Fokker-Planck theory to describe (passive) nuclear transport and could therefore potentially be of high interest. However, the choice of NPC scaffold and corresponding FG-Nups leads to an underprediction of the actual FG-Nup concentration, which affects the reported

readouts and limits the impact of the results for the wider NPC community. The authors should properly remedy this before the paper can be accepted for publication in Nature Communications.

Our response: We thank the Reviewer for the careful review of our work and for the constructive criticism that has prompted us to repeat our calculation on another, more recent model of the NPC.

Major concerns:

1. *(1a) The authors refer to the used NPC model as a ‘complete’ NPC structure. However, the used structure (from Lin/Hoelz) is a ‘composite’ NPC consisting of C. thermophilum Nucleoporin complexes fit into a cryo-EM map of the human NPC. The authors here use a set of five intrinsically disordered FG-Nups from the NPC central channel of C. thermophilum, omitting several cytoplasmic and nucleoplasmic FG-Nups. As a result, the real FG-Nup mass concentration is underestimated and the spatial distribution of the FG-Nups inside the NPC is considerably affected. This will strongly influence the reported readouts (PMF, transport times, etc.) as these are all based on the concentration and organization of the FG-Nups inside the channel.*

Response 1a. First, we agree with the reviewer that the model is better described as ‘composite’ than ‘complete,’ and change the adjective in the main text. For reference, we reproduce below the panels A and B of Figure 1 from [Lin2016] which schematically show the categories of nups present in their model and which species of nup are present. In panel B, resolved portions are overlined in black, and rectangles classify sections of the amino-acid sequences. For our coarse-grained simulations, we included the FG repeat sections of (Nup57, Nup49, Nsp1, Nic96, Nup145N). Among other adaptor nups and coat nups, the unresolved portions of the nups are either helical (red) or unstructured linker (gray), which are proposed to primarily link the nups together within the scaffold. [Lin2016] classifies Nup159 as a cytoplasmic filament nup, and does not include the short resolved portion in their published “composite symmetric core.” Therefore, although it contains an FG repeat section, we chose to exclude Nup159 from our CG model.

Response Fig. 4: NPC schematic and nup region classification; Figure 1A, B from [Lin2016].

Changes made to the manuscript:

Page 26: “Note, the unresolved portions of other adaptor nups and coat nups present in the Lin2016 structure are either helical or unstructured linker, which are proposed to primarily link the nups together within the scaffold. Ref 45 classifies Nup159 as a cytoplasmic filament nup, and does not include the short resolved portion in their published composite symmetric core. Therefore, although Nup159 contains an FG-repeat section, we chose to exclude that nup species from our CG model.”

Response 1a continued. We also agree with the reviewer that a greater FG-nup mass density will influence the reported readouts, including PMF and mean first-passage time. To determine what effect a higher FG-nup density will have on our results, we have built and simulated a CG model of another published structure constructed by Kim et al. [Kim2018] (referred to as the Kim2018 structure), a yeast NPC structure from the Rout and Sali laboratories. The resulting CG simulation trajectories of the Kim2018 structure were analyzed using our void analysis / Fokker-Planck method. The resulting local FG-nup densities, PMFs, MFPTs and percolation threshold are now included in the revised manuscript as Figure 6 (reproduced below). As you can see, the FG-nup mesh of the yeast Kim2018 NPC structure is about 15% denser than that of the composite Lin2016 structure, which make PMF barriers a bit higher, shifts the percolation threshold to slightly smaller probe radius and makes MFPT values a bit larger. But, despite these small quantitative changes, we still clearly see a soft-to-hard barrier change coinciding with the percolation

transition. These new results confirm the robustness of our conclusions. We are grateful to the reviewer for prompting us to do these additional simulations.

Figure 6: Passive diffusion of proteins through yeast NPC.

Changes made to the manuscript: We added new main text Figure 6, new “Passive transport through yeast NPC” section of the main text (page 21-24), updated Methods to describe procedures used to build and run a CG model of the Kim2018 structure.

1b. For example, the reported densities in the NPC central channel are quite different from earlier reported values (e.g. Ghavami [1], Huang [2], Peyro [3]). These differences should be discussed in light of the different stoichiometries of the composite NPC (C. thermophilum versus human/yeast): there are 48 copies of Nsp1 and Nup62 in yeast for instance, not 32, and also Nup98 (human) and Nup100+116+145N (yeast) have 48 copies rather than 32. Furthermore, no cytoplasmic and nucleoplasmic FG-Nups are accounted for, which are known to play an important role in establishing the permeability barrier.

Response 1b. Thank you for the suggestion to compare the average nup density in our work to [Ghavami2014, Peyro2015, Huang2019]. First we note that Ghavami et al. [Ghavami2014] and Peyro et al. [Peyro2015] appear to use the same model for the entire NPC, so we will focus our comparison on the density reported by [Ghavami2014]. Our new Supplementary Fig. 1a (reproduced below) provides a to-scale side-by-side comparison of the average nup density from [Ghavami2014] (Figure 4A of that study) and from [Huang2019] (Figure 1C of that study) to densities resulting from our CG simulations of the Lin2016 (composite structure) and Kim2018 (yeast NPC) models. To enable such quantitative comparison, we converted the FG-nup number density units used in

[Ghavami2014] and the amino acid concentration units used in [Huang2019] to mg/mL assuming that each amino acid residue has a molecular weight of 120 Dalton.

In panel b of this new Supplementary Fig. 1, we provide a comparison of the scaffold, anchor positions and nup stoichiometries for the same four models, two from our work. Our model based on [Lin2016] has 32 Nsp1 segments present, while [Huang2019] and our model based on [Kim2018] both have 48. Furthermore, [Huang2019] and [Kim2018] have 48 from the combination of Nup100+116+145N, whereas [Lin2016] has 32 of only Nup145N. These differences explain, in part, why we observe a slightly lower nup density in the middle of the central channel for our [Lin2016] model, compared to either [Huang2019] or our [Kim2018] model.

Supplementary Fig. 1: Density and stoichiometry of FG-nups across recent computational models of NPC.

Changes made to the manuscript::

Added new Supplementary Figure 1 providing side-by-side comparison of the FG-nup densities and of the nup stoichiometries.

Page 7: “In comparison to earlier computational studies of model NPC systems [49,50], Supplementary Fig. 1a, our local FG-nup density map has similar values in the middle of the central channel but shows much lower values near the anchor points, which we attribute to differences in both the scaffold dimensions and the nups’ stoichiometry. Our density map is

comparable to that reported in a more recent computation study [35], Supplementary Fig. 1a. We find the spatial distribution of the hydrophobic residues of the FG-nups, Supplementary Fig. 2d, to follow the overall FG-nup density, similar to observations reported in Ref. 49.”

Page 21: “In accordance with the composition of the Kim2018 structure (which lacks symmetry with respect to the NPC’s midplane), the resulting average density of the FG-nups has an asymmetric shape, Fig. 6b, but spans a similar range of density values as in the Lin2016 model, Fig. 1b. The average density of the FG-nup amino acids within the central region of the mesh (a cylinder of 30 nm diameter and 30 nm length centered at the origin) was 68.8 ± 2.7 mg/mL for the Kim2018 model, slightly higher than in the Lin2016 model, 59.8 ± 3.0 mg/mL, where the standard deviation reflects temporal fluctuations when sampled every $0.1 \mu\text{s}$.”

1.c Nup145N is a homolog of human Nup98 and yeast Nup100/116/145N, so one would expect a similar behavior. However, Nup145N seems to have little interactions with the other central channel FG-Nups (Fig. 1c and 5g) in contrast to yeast for instance where Nup100/116/145N collectively form a dense ring-shaped region, which plays an important role in controlling passive diffusion. The authors should address these discrepancies. For instance, what is the average charge over hydrophobicity ratio (C/H) of the FG-Nups in the current study and how does this compare with the homolog-structures in yeast and human NPCs? In addition, Nup145N is anchored far towards the cytoplasmic and nucleoplasmic side (Fig. 5g) and not centrally anchored like the human Nup98 and yeast Nup100/116/145N homologs. All these aspects should be discussed as well as these likely play an important role in the distribution of the FG-Nups and thus the readouts of the void model and the Fokker-Planck analysis.

Response 1c. In our model based on the Lin2016 NPC scaffold, FG-nup domains have the following C/H ratios: Nup145N (0.121), Nic96 (0.078), Nsp1 (0.097), Nup49 (0.070), Nup57 (0.061). Including all residues present in the Lin2016 model, $C/H \sim 0.102$, i.e. about ten times as many hydrophobic residues as charged ones. When comparing *C. thermophilum* Nup145N to human Nup98 and yeast Nup100/116/145N, we obtain the following results: G0SAK3 *C. thermo.* Nup145N residues 1-732 (0.121); P52948 Human Nup98 residues 1-500 (0.107); Q02629 Yeast Nup100 residues 1-794 (0.059); Q02630 Yeast Nup116 residues 1-939 (0.085); P49687 Yeast Nup145N residues 1-242 (0.092). From this comparison, we observe *C. thermo.* Nup145N (from Lin2016), and its homologs Human Nup98 and Yeast Nup100/116/145N (from Kim2018) all have relatively close C/H ratios, from 0.06 to 0.12.

We added new panels to Supplementary Fig. 2 (below) to provide the charge to hydrophobicity ratio of all FG-nup segments present in the composite Lin2016 NPC model and Kim2018 yeast NPC model. Nup145N in our Lin2016 NPC model has a C/H ratio of 0.121, which is close to homologous human Nup98 (0.107) and yeast Nup100 (0.059), Nup116 (0.085) and Nup145N (0.092).

As seen in Supplementary Fig. 1a, the more biologically relevant Kim2018 yeast NPC model does indeed have a slightly higher overall density within its central channel than our composite Lin2016 NPC model. We attribute that difference in density, in large part, to the fact that the Kim2018 NPC model contains 48 copies from Nup100+116+145N, whereas the Lin2016 NPC model only contains 32 copies of Nup145N (anchored more externally), overall stoichiometries provided in Supplementary Fig. 1b.

Supplementary Fig. 2: Properties of the FG-nup mesh.

Changes made to the manuscript:

Provided additional analysis of C/H ratio as panels to Supplementary Figure 2, along with a descriptive caption that summarizes the above discussion.

Page 23: “We attribute the difference in density, in large part, to the fact that the Kim2018 model contains 48 copies from Nup100+116+145N, whereas the Lin2016 NPC model only contains 32 copies of Nup145N (anchored more externally), see Supplementary Fig. 1 for the stoichiometry of each structure.”

1d. The authors find that most of the interchain contacts were formed between a hydrophobic and a polar residue (45%), about twice as likely as a hydrophobic-hydrophobic or polar-polar contact. This is in contrast to what is reported in many other studies, both experimental and theoretical, and again raises the question whether the stoichiometry of the composite NPC structure used is biologically relevant.

Response 1d. Indeed, many previous numerical studies have concluded that hydrophobic residues are important, for example, by mutating hydrophobic residues to polar serine and observing a major change in the overall FG-nup density. In terms of contacts specifically, we would like to point out that, for Supplementary Fig. 2b and 2f, contacts involving hydrophobic residues (HH, HP, HC) make up over 60% of all contacts present. In our literature search, we did not find contact analysis done in the same pairwise manner presented above. While a hydrophobic-polar pair is the most likely for both our NPC CG simulations, hydrophobic residues contribute the most to inter-nup contacts overall compared to either charged or polar residues.

For reference, below we provide Figure 3 from [Ghavami2014], which graphically visualizes the densities due to all FG-nup residues (orange), charged residues (red) and hydrophobic residues (green).

Response Fig. 5 (same as Response Fig. 1): The three-dimensional density distribution of different amino acids inside the NPC (Figure 3 from [Ghavami2014]). Density of all FG-nup residues is shown in orange, charged residues in red and hydrophobic residues in green.

Our density maps exhibit a qualitatively similar property, in that the volume occupied by hydrophobic residues spans a similar space to all FG-nup residues at a lower overall

density. From our results, we can compare Fig. 1b (all for Lin2016) to Supplementary Fig. 2d (hydrophobic for Lin2016), as well as Fig. 6b (all for Kim2018) to Supplementary Fig. 2h (hydrophobic for Kim2018). From our simulations, all residues reach an upper density limit of about 130 mg/mL whereas hydrophobic residues alone reach about 65 mg/mL, but span a similar overall volume.

Changes made to the manuscript:

Pages 23: “Contact analysis of the Kim2018 equilibration trajectory shows that, just as with the Lin2016 model, a hydrophobic-polar pairing is more likely than a hydrophobic-hydrophobic or polar-polar pairing, Supplementary Fig. 2. We would like to point out that the contacts that involve hydrophobic residues make up over 60% of all contacts observed over the course of CG simulations of both models. And although we find a hydrophobic-polar residue contact to be the most prevalent, the hydrophobic residues contribute, overall, the largest fraction of inter-nup contacts, in comparison to either charged or polar residues.”

- 2. In recent years it has become undeniably evident that, at all times, a significant concentration of nuclear transport receptors is present in the NPC (see e.g. work from the groups of Lim and Rout). Considering that a large part of the paper depends on PMF curves obtained from available voids, this potentially has a drastic effect on the observed transport times. This is ignored in the current paper.*

Our response. Indeed, experimental studies suggest that up to 100 nuclear transport receptors (NTR) can be located in or around the NPC *in vivo* [Lowe2015], with 50 copies being a reasonable estimate at physiological conditions. While a systematic study of active transport through the NPC is beyond the scope of this manuscript, we can estimate the effect of NTRs on the percolation threshold. Taking importin- β of 95 kDa molecular mass as a typical representative, we estimate its volume to be around 172 nm³. The volume of the NPC’s central mesh that is 25 nm in radius and 40 nm in height is 78500 nm³. Taking the worst case scenario of all 50 NTR present within the central mesh of the NPC, we find the fraction of the NPC volume excluded by the presence of 50 NTRs is thus $50 \cdot 172 / 78500$ about 11%. Thus, the volume available to the FG-nup mesh will be reduced by 11%, which will make the mesh about 11% more dense. This difference is comparable to the difference between our Lin2016 and Kim2018 models. Thus, while we expect the presence of the NTR to push the percolation threshold (and hence the soft-to-hard barrier transition) to smaller protein sizes, the difference is not expected to be dramatic and the overall conclusions of the study hold.

Changes made to the manuscript:

Page 24: “Conversely, a presence of 50 nuclear transport factors, such as importin- β , within the central mesh of the NPC [57], is expected to increase the effective FG-nup density by approximately 11%, pushing the percolation threshold (and hence the soft-to-hard barrier transition) to smaller protein sizes.”

3. *The current study predicts a more-or-less linear dependence of the transport times (MFPT) with the mass of the cargo proteins. Experimental data from the groups of Veenhoff and Rout (refs. 6 and 8 from the paper, resp.), however, show a cubic dependence of the transport times on the molecular mass. How can these seemingly contrasting observations be justified?*

Response Fig. 6: Mean first-passage time versus molecular mass measured from experiment (orange) and simulation (blue); Figure 4B from [Timney2016].

Our response: Yes, they can be!

For reference, we reproduce above Figure 4B from [Timney2016]. Light blue dots show scaled simulation results from that study. Orange data points indicate experimental data

from [Timney2016] (diamonds) and [Popken2015] (triangles). Lines of light blue and orange are power-law fits to the simulation and combined experimental datasets, respectively.

Below, we reproduce our updated Supplementary Fig. 14, where we directly compare the results of previous studies to the results of our simulations. Specifically, we plot the mean first-passage time (MFPT) vs protein mass as reported in Timney2016 and Popken2015 along with the MFPT resulting from our CG (dark blue) and void analysis / Fokker-Planck calculations (red). Note that both x and y axes are logarithmic. To make direct quantitative comparison, the data from the three datasets ([Timney2016] (simulation), [Timney2016] (experiment) and [Popken2015] (experiment)) were all multiplied by a single constant value: the ratio of the hemoglobin's (61.5 kDa) MFPT obtained from our CG simulations and the MFPT reported by [Timney2016] simulations for a sphere of a 61 kDa mass. The gray shaded rectangle in the figure indicates where we observe a transition region from power-law to exponential scaling, based on our Fokker-Planck calculations of MFPT.

Supplementary Fig. 14: Comparison to results of previous studies of passive diffusion.

From this direct comparison, we observe that results from our study's Fokker-Planck calculations are in relatively good agreement with data from [Timney2016] (sim), [Timney2016] (exp) and [Popken2015] (exp) for a range of protein mass from about 45kDa

to 200kDa. From 25kDa to 45kDa, we observe simulation and experimental MFPT values from [Timney2016] to be shorter than MFPTs calculated from our coarse-grained simulations or Fokker-Planck approach. Experimental and simulation results from [Timney2016] did not extend lower than 25kDa in protein mass. The discrepancy in scaling relations based on our CG simulations (linear) versus scaling observed in the Veenhoff and Rout studies (cubic) is due, in part, to the ranges in mass considered for such fitting: ~5kDa to ~100kDa for our CG simulations vs ~25kDa to ~200kDa for the combined data from Veenhoff and Rout studies.

As we describe in detail in our response to comment 4 of Reviewer 1, a single power-law fit cannot describe the behavior of MFPT in the entire range of protein mass. Furthermore, the intermediate region, where MFPT switches from power-law to exponential scaling, as determined by our Fokker-Planck results, spans from about 25kDa to 90kDa, which is a significant fraction of the range considered by [Timney2016], which may indeed be approximated by a cubic dependence.

Changes made to the manuscript:

We have updated what is now Supplementary Figure 14 to provide direct comparison to previous data.

Page 17: “For our Lin2016 NPC model both with and without FG-nups present, the MFPTs computed using our Fokker-Planck approach are in excellent agreement with the CG simulation data for both wide, Fig. 5a,b, and narrow, Supplementary Fig. 13a,b, confinement potentials, as well as with the simulation and experimental results of previous studies, when scaled by the MFPT of hemoglobin, Supplementary Fig. 14.”

We have also expanded our description of the fitting procedures on Page 19 to explain how a power law dependence can emerge from an exponential one through Taylor expansion in the limit of small barrier height.

Minor concerns:

Some additional concerns are listed below:

(a) The authors report simulation times using an estimated effective timestep. The used time scaling is justified by the claim that the FG-Nups are fully extended IDPs. However, the FG-Nups (especially in the NPC central channel) are not generally considered as fully extended IDPs (see the work of Yamada et al. 2010), and thus the used timestep rescaling might not be appropriate.

Our response: Thanks for pointing this out. Indeed, we used the adjective “fully” somewhat carelessly in the description of the internal friction and the original study [Soranno2012] that quantified the internal friction of IDPs did not use this term. The results in that study have shown that chemically denatured IDPs (by GdmCl) exhibit low internal friction, which was inferred from FRET-measured end-to-end relaxation times. The natural question, then, is what does “extended” mean in this context? The inset of Fig 5A of that study shows that the internal friction drops to very small values for three peptides 55, 67 and 111 amino acids long when the radius of gyration increased above ~ 2.3 , ~ 2.4 , and ~ 3.2 nm, respectively, see reproduction of the inset below.

Response Fig. 6: Dependence of internal friction on radius of gyration for three peptides in GdmCl solution. Adapted from Fig. 5A inset of [Soranno2012].

To determine the scaling regime seen for FG-nups in our simulations, we plotted (as blue circles) the average R_g values as a function of the FG-nup length (see new Supplementary Fig. 16 below). For reference, we also plotted the data from [Soranno2012] (as magenta circles). Both curves were found to be well approximated by the following scaling relation for denatured proteins: ($R_g(N) = 2.02 \cdot N^{0.6}$; [Ding2005]). In the same plot, we also show the scaling law expected for globular proteins ($R_g(N) = 2.2 \cdot N^{0.38}$; [Skolnick1997]). Hence, it appears that the FG-nups in our model are sufficiently extended that internal friction can be neglected. To avoid confusion, we have removed the adjective “fully” from the main text.

In addition, we clarify in the revised text that hydrodynamic interactions, which would scale the relaxation times non-uniformly, are expected to be negligible due to screening in a dense mesh as noted by DeGennes and more recently in the context of the NPC by Moussavi-Bagyi.

Supplementary Fig. 16: Scaling behavior of the FG-nups radius of gyration.

Changes made to the manuscript:

Added new Supplementary Figure 16 (reproduced above) detailing scaling of the FG-nups' radii of gyration as a function of the FG-nups' length.

Pages 29-30: "We chose to report our simulation results using the effective time step because we found the average radius of gyration of the FG-nups in our Lin2016 model to scale with the FG-nup length following a scaling law associated with extended intrinsically disordered peptides, Supplementary Fig. 16, which exhibit low internal friction [64]. Furthermore, hydrodynamic interactions are expected to be negligible due to screening in a dense mesh [65]."

(b) More details should be given on the implementation of the coarse-grained model. The authors write: '... the beads were connected into a polymer chain via a harmonic spring potential, see Refs. 49 and 50 for details'. From this it is not clear whether the angular and dihedral potentials (originally present in the CG model) are included or not.

Our response: Thank you for pointing out this omission. Yes, angular and dihedral potentials were also included.

Changes made to the manuscript:

Pages 26-27: "Each amino acid residue of the fragments was represented by one CG bead; the beads were connected into a polymer chain via harmonic spring potentials describing the bonds, angles and dihedral angles terms, see Refs. 48 and 49 for details."

(c) The authors have verified the implementation of the coarse-grained model from a brush simulation. However, it is not clear how the height is determined, since this is not trivial for brushes. It would also be good to do a check of the Stokes' radii of several yeast FG-Nups to assess whether single-chain properties are correctly retrieved (see the work of Yamada et al. 2010).

Our response: In response to this comment, we now provide more information about our validation simulations. To create the brush, Nup62 fragments (residues 1-240) were grafted onto a flat surface in a 10x10 array, each anchor point 2.4 nm apart. The residue in the entirety of the brush farthest from the flat surface for a specific frame was used to define the “maximum brush height” for that simulation step. The figure below provides side-by-side comparison of our simulations (panel b) to the Figure S1 from [Ghavami2014].

Response Fig. 7: Maximum Nup62 brush height as a function of simulation steps. a, The inset of Supporting Figure S1 in [Ghavami2014]. b, Maximum brush height measured from a simulation using our implementation of the same coarse-grained force field.

Ghavami et al. [Ghavami2014] also made a direct comparison of the Stokes radius predicted from their simulations of individual yeast FG-nups to experimental values from [Yamada2010], as shown in Supporting Material Table S3 (below).

FG-nup segment*	length	$R_{s, \text{experiment}}$	$R_{s, \text{predict}}$	Errors %
Nsp1n_lc (AA 1-172) [†]	172	27.1	32.28	16.05
Nup116m_lc (AA 165-715)	551	46.5	45.97	1.15
Nup100n_lc (AA 2-610)	625	48.7	49.43	1.48
Nup49_lc (AA 1-215)	215	26.9	33.52	19.74
Nup42_lc (AA 1-212)	212	28.4	29.29	3.04
Nup57_lc (AA 1-255)	255	31.9	34.66	7.97
Nup145n_lc (AA 1-242)	242	28.2	32.22	12.47
Nup1c_lc (AA 798-1078)	279	32.4	36.31	10.77
Nup159_hc (AA 441-881)	441	55.4	61.13	9.37
Nup60_hc (AA 389-539)	151	31.3	34.81	10.09
Nup1m_hc (AA 220-797)	578	67.9	72.73	6.64
Nup2_hc (AA 186-561)	376	59.8	55.84	7.1
Nsp1m_hc (AA 173-603)	431	65.3	64.95	0.54
Nup145ns (AA 243-433)	191	29.8	36.47	18.29
Nup100s (AA 611-800)	190	36.6	40.91	10.53
Nup116s (AA 765-960)	196	39.1	42.28	7.52

* "lc" stands for low charge, "hc" stands for high charge content and "s" refers to the stalk region of the Nup (12).

[†] In the experiments of Yamada et al. (12), some FG-nup segments have a tag of several Histidines at the end of the Nup. To investigate this, we have simulated Nsp1n_lc with 6 charged Histidine amino-acids added at the N-terminus (this is in line with the FG-nup sequence studied by Yamada et al.). We have chosen Nsp1n_lc because it is the shortest, low-charge segment, so if protonation would play a role, it would have a maximal effect on this segment. The calculated Stokes radius was found to increase by a small amount, from 32.28 to 33.45, an increase 3.4%.

TABLE S3 The predicted R_s values for FG-nup segments compared to the experimental values (12)

Response Table 1: Stokes Radii of FG-nup segments simulated in isolation compared to experimental measurements performed by [Yamada2010].

We selected three FG-nups from this table and simulated them in isolation using our implementation of the CG force field presented in [Ghavami2014]: Nsp1n_lc, Nup100s and Nup60_hc. Our simulation data showed good agreement with those of [Ghavami2014] (see panel a in the Supplementary Fig. 15 reproduced below), which in turn matched the experimental data of [Yamada2010].

Supplementary Fig. 15: Validation of the coarse-grained model implementation.

Changes made to the manuscript:

Added new Supplementary Figure 15 (reproduced above) detailing our validation of the CG force field implementation.

Page 27: “We validated our implementation of the CG model by measuring the Stokes radii of three FG-nup species in isolation and the maximum height of a 10×10 Nup62 brush, Supplementary Fig. 15, which reproduced the values reported in Ref. 49.”

[1] Ali Ghavami, Liesbeth M. Veenhoff, Erik van der Giessen, Patrick R. Onck, Probing the Disordered Domain of the Nuclear Pore Complex through Coarse-Grained Molecular Dynamics Simulations. *Biophysical Journal* 107, 2014, p. 1393-1402. DOI: 10.1016/j.bpj.2014.07.060

[2] Kai Huang, Mario Tagliazucchi, Sung Hyun Park, Yitzhak Rabin, Igal Szleifer, Nanocompartmentalization of the Nuclear Pore Lumen. *Biophysical Journal* 118, 2019, p. 219-231. DOI: 10.1016/j.bpj.2019.11.024

[3] Mohaddeseh Peyro, Mohammad Soheilypour, Ali Ghavami, Mohammad R. K. Mofrad, Nucleoporin's Like Charge Regions Are Major Regulators of FG Coverage and Dynamics Inside the Nuclear Pore Complex. *PLoS ONE* 10(12): e0143745. doi:10.1371/journal.pone.0143745

References used in the response to Reviewer 2:

[Lin2016] = Lin, Daniel H., Tobias Stuwe, Sandra Schilbach, Emily J. Rundlet, Thibaud Perriches, George Mobbs, Yanbin Fan et al. "Architecture of the symmetric core of the nuclear pore." *Science* 352, no. 6283 (2016): aaf1015.

[Kim2018] = Kim, Seung Joong, Javier Fernandez-Martinez, Ilona Nudelman, Yi Shi, Wenzhu Zhang, Barak Raveh, Thurston Herricks et al. "Integrative structure and functional anatomy of a nuclear pore complex." *Nature* 555, no. 7697 (2018): 475-482.

[Ghavami2014] = Ghavami, Ali, Liesbeth M. Veenhoff, Erik van der Giessen, and Patrick R. Onck. "Probing the disordered domain of the nuclear pore complex through coarse-grained molecular dynamics simulations." *Biophysical journal* 107, no. 6 (2014): 1393-1402.

[Huang2019] = Huang, Kai, Mario Tagliazucchi, Sung Hyun Park, Yitzhak Rabin, and Igal Szleifer. "Nanocompartmentalization of the nuclear pore lumen." *Biophysical journal* 118, no. 1 (2020): 219-231.

[Peyro2015] = Peyro, Mohaddeseh, Mohammad Soheilypour, Ali Ghavami, and Mohammad RK Mofrad. "Nucleoporin's like charge regions are major regulators of FG coverage and dynamics inside the nuclear pore complex." *PLoS One* 10, no. 12 (2015): e0143745.

[Lowe2015] = Lowe, Alan R., Jeffrey H. Tang, Jaime Yassif, Michael Graf, William YC Huang, Jay T. Groves, Karsten Weis, and Jan T. Liphardt. "Importin- β modulates the permeability of the nuclear pore complex in a Ran-dependent manner." *Elife* 4 (2015): e04052.

[Timney2016] = Timney, Benjamin L., Barak Raveh, Roxana Mironska, Jill M. Trivedi, Seung Joong Kim, Daniel Russel, Susan R. Wentz, Andrej Sali, and Michael P. Rout. "Simple rules for passive diffusion through the nuclear pore complex." *Journal of Cell Biology* 215, no. 1 (2016): 57-76.

[Popken2015] = Popken, Petra, Ali Ghavami, Patrick R. Onck, Bert Poolman, and Liesbeth M. Veenhoff. "Size-dependent leak of soluble and membrane proteins through the yeast nuclear pore complex." *Molecular biology of the cell* 26, no. 7 (2015): 1386-1394.

[Soranno2012] = Soranno, Andrea, Brigitte Buchli, Daniel Nettels, Ryan R. Cheng, Sonja Müller-Späth, Shawn H. Pfeil, Armin Hoffmann, Everett A. Lipman, Dmitrii E. Makarov, and Benjamin Schuler. "Quantifying internal friction in unfolded and intrinsically disordered proteins with single-molecule spectroscopy." *Proceedings of the National Academy of Sciences* 109, no. 44 (2012): 17800-17806.

[Ding2005] = Ding, Feng, Ramesh K. Jha, and Nikolay V. Dokholyan. "Scaling behavior and structure of denatured proteins." *Structure* 13, no. 7 (2005): 1047-1054.

[Skolnick1997] = Skolnick, Jeffrey, Andrzej Kolinski, and Angel R. Ortiz. "MONSSTER: a method for folding globular proteins with a small number of distance restraints." *Journal of molecular biology* 265, no. 2 (1997): 217-241.

[Yamada2010] = Yamada, Justin, Joshua L. Phillips, Samir Patel, Gabriel Goldfien, Alison Calestagne-Morelli, Hans Huang, Ryan Reza et al. "A bimodal distribution of two distinct categories of intrinsically disordered structures with separate functions in FG nucleoporins." *Molecular & Cellular Proteomics* 9, no. 10 (2010): 2205-2224.

Reviewer #3 (Remarks to the Author):

The authors perform extensive large-scale coarse-grained (CG) Molecular Dynamics (actually Brownian Dynamics) simulations on the routes and rate of transport of several protein molecules through Nuclear Pore Complex (NPC). The existence of a polymer mesh made of natively unstructured proteins (nucleoporins) that are grafted to the pore scaffold forces the translocating protein to find its way through holes in the mesh. This becomes more difficult to do as the protein size increases, for reasons elucidated in the manuscript, resulting in a “hard” (virtually impenetrable) barrier for proteins above a certain size. All of this is interesting enough from a mechanistic point of view, and nicely supported by an approximate theoretical model that gives results in surprisingly close agreement with the full CG simulations.

In general, unassisted (“passive”) diffusion of relatively small proteins is not what makes the NPC interesting. In fact, large proteins and strands of mRNA can translocate through the NPC, but only if they attach themselves to special globular receptor proteins, sometimes called Nuclear Transport Factors (NTFs). These NTFs have hydrophobic surfaces that bind transiently to segments of the polymer mesh (i.e., nucleoporin) proteins. Without such binding events, the NPC does not allow for selective transport of large protein or RNA cargos, which is its critical physiological function. Such transient binding effects are not considered in the authors’ model, as they acknowledge in the Conclusion section.

So, this paper does not resolve the fundamental mystery of how the NPC operates, namely, how it enables the transport of large cargo molecules in/out of the nucleus in a highly selective and highly efficient (rate of translocation on the order of 1 cargo molecule/ms) manner. Nevertheless, it presents a carefully crafted study of passive diffusion through the NPC, which provides novel insights into the mechanism of this process.

Our response: We thank the reviewer for the thorough review of our work and for the comments that have helped us clarify assumptions that went into constructing our computational model of the NPC

The following questions should be answered prior to publication:

1) Where does Importin- β (a representative member of the karyopherin [NTF] receptor family) fall on this protein size range considered in the present work (e.g., in Fig. 5d). Presumably, it is above the cutoff size for no passive transport.

Our response: Importin- β is about 95kDa, which is in the “hard barrier” scaling region (90kDa and above). We’ve added a sentence in the main text to state that, by size alone, this representative nuclear transport factor is in the hard, exponential scaling regime.

Changes made to the manuscript:

Page 20 of the main text: “Note that importin- β (95 kDa), a representative karyopherin (i.e. nuclear transport) receptor, by its size alone, is expected to experience considerable reduction of spontaneous transport because of the presence of the FG-nup mesh.”

2) Diffusion constant $D(z)$ profile: Explain Eq. 17, namely, the dependence on cross sectional area at point z on the pore axis. Where does this functional form come from? If there is no rigorous justification for this profile, how sensitive are the results obtained for Mean First Passage Times (MFPTs), for example, to the functional form employed here? This question is relevant since the authors claim quantitative success of this 1-d drift-diffusion model in explaining their many-atom CG BD simulation data.

Our response: We thank the referees for this insightful comment. In our calculations of the MFPT, we used a local diffusion model introduced by Reguera and J. M. Rubi [Phys. Rev. E, **64** 061106 (2001)]. Previously, Zwanzig [J. Phys. Chem. **96**, 3926 (1992)] analytically derived an approximate 1D kinetic equation from a Smoluchowski equation to describe diffusion in a 2D/3D tube and justified the usage of a position-dependent equivalent diffusion coefficient. Following that derivation, Reguera and Rubi [Phys. Rev. E, **64** 061106 (2001)] described how such equivalent diffusion coefficient should change in the case of a tube of varying cross section. We chose the local diffusion expression from Reguera’s study because it better approximates diffusion through the void volumes within the NPC if compared to the Zwanzig’s expression. We also justified our choice by comparing the MFPT obtained from Fokker-Planck calculations to the values directly extracted from our CG MD simulation.

In response to the reviewer’s comment, we repeated our calculations of the MFPT using another profile of the local diffusion coefficient, the one originally introduced by Zwanzig. The figure below compares the results of the two calculations, where data obtained using the functional form from [Reguera2001] are shown in orange and from [Zwanzig1992] in purple. The resulting MFPT values are very close to each other, indicating that our Fokker-Planck calculations are not sensitive to the functional form of $D(z)$.

Supplementary Fig. 18: Effect of local diffusion model on Fokker-Planck MFPT.

Changes made to the manuscript:

Added the above figure as Supplementary Fig. 18 to the Supplementary Information.

Added the following description to Methods (Page 38): “The results of the Fokker-Planck MFPT analysis displayed in Fig. 5 and Fig. 6 were obtained using the Reguera and Rubi model of the local diffusion coefficient, Eq. 18 above. To check if our results are sensitive to this particular choice of the local diffusion model, we have repeated our Fokker-Planck MFPT calculations using the local diffusion model of Zwanzig, [Equation] where D_0 and $w(z)$ have the same meaning as above. We found the choice of the local diffusion model to have little influence on the results of our MFPT analysis, Supplementary Fig. 18.”

3) *Potential of Mean Force (PMF) calculation via “Boltzmann inversion” (p. 10): I presume the CG BD trajectory runs long enough for the protein molecule to cross through the pore multiple times, and spend significant amounts of time in the reservoirs on either side of the pore. Please confirm.*

Our response: Yes, all proteins other than the largest (GAPDH) one were observed to cross through the NPC multiple times and spend significant amounts of time in either reservoirs. We describe this fact early on when introducing results of our CG simulations (Page 9 of the main text): “In a typical simulation, a protein was observed to translocate from one side of the NPC to the other multiple times as the FG-nup mesh continuously changed its conformation, see Supplementary Movie 2.”

The quality of CG simulation sampling and, hence, of the Boltzmann inversion procedures, can be evaluated from the amount of asymmetry in the raw, non-symmetrized $P(z)$ and $PMF(z)$ curves, which we provided for that very specific reason as Supplementary Figs. 4-7. We would also like to point out that the void analysis-generated PMF is better sampled than a brute-force CG simulation, because the rigid-body protein occupies one position per frame while one frame's FG-nups configuration can provide a $P(z)$ profile along the entire pore axis.

Changes made to the manuscript:

Page 11: “The amount of asymmetry observed for the raw, non-symmetrized $P(z)$ and $PMF(z)$ curves, Supplementary Figs. 4–7, provides a sense of the sampling quality.”

4) Usage of the cylindrical confining potential. This is done to accelerate the calculation of MFPTs and transport rate constants by reducing the amount of the time the protein bounces around in the reservoirs. Fine, but, particularly for calculation of experimentally observable properties like the transport rate, this confinement potential should not affect the predicted results. Has this been confirmed, presumably by increasing the value of the confinement radius R_{con} until the extracted rate constant does not change with further increase of R_{con} ?

Our response: Yes, the confinement radius determines the effective concentration of the protein, which affects the MFPT by affecting the “attempt rate”. When changing the radius of the confinement potential from 25 to 50 nm, the overall rate of transport decreases by about a factor of 3, which is smaller than the expected factor of 4 because the protein scaffold occupies a non-negligible fraction of the confinement potential volume. When making comparison to experimental results (on Page 12), we used data obtained using the 50 nm confinement potential as we expect it to better represent the case where such confinement potential is absent, as in experiment.

At the same time, the distributions of the “crossing time” (Figure 3e) should not depend on the confinement potential radius, as it is a property of the mass. Indeed, we find that to be the case by plotting in the figure below the distributions of crossing times separately for $R = 25\text{nm}$ (color-filled), $R = 50\text{nm}$ (black outline).

Response Fig. 8: Distributions of crossing times for successful coarse-grained simulation translocation events, confinement $R = 25\text{ nm}$ (filled) and $R = 50\text{ nm}$ (empty) plotted separately.

Changes made to the manuscript:

Page 14 "...crossing time does not depend on the width of the confinement potential"

Caption to Supplementary Fig. 3: “Compared to the 50 nm-radius confinement data, Fig. 2h, the MFPT values are a factor of 3 higher because of the higher effective concentration of the protein.”

References used in the response to Reviewer 3:

[Reguera2001] = Reguera, David, and J. M. Rubi. "Kinetic equations for diffusion in the presence of entropic barriers." *Physical Review E* 64, no. 6 (2001): 061106.

[Zwanzig1992] = Zwanzig, Robert. "Diffusion past an entropy barrier." *The Journal of Physical Chemistry* 96, no. 10 (1992): 3926-3930.

Reviewers' Comments:

Reviewer #1:

Remarks to the Author:

The authors have answered all my requests. In particular the comparison with the state of the art on nuclear pore modelling have been strengthened which highlights the novelty of this work for experimentalists. I think this new version of the manuscript will be a valuable tool for the understanding of nucleo-cytoplasmic transport.

For these reasons I strongly support the publication of this work in Nature Communications

Reviewer #2:

Remarks to the Author:

Below is our review of the revised manuscript of NCOMMS-21-49443A: Percolation transition determines protein size limit for passive transport through the nuclear pore complex

The authors have added new results on a yeast NPC structure (Kim et al) to answer our main concerns on the low FG-Nup mass in the initial composite structure (Lin et al). However, there are several issues with the assumptions made in this new computational model related to mass inclusion and the performed contact analysis. The authors should properly remedy these before the paper can be accepted for publication.

The review below is numbered according to the answers the authors gave (named 'Response 1a, 1b, etc.') to the initial review (reviewer #2) in their rebuttal letter.

****Response 1a)**

We thank the authors for taking over our suggestion to describe the Lin2016 model as 'composite' rather than 'complete'.

****Response 1b)**

The authors have tried to address our concerns about their reported NPC densities by repeating their analyses on a second, more complete NPC model (the Kim2018 model). However, we have several remarks on the added section that discusses the Kim2018 coarse-grained structure:

A. Based on the supplied FG-Nup sequence lengths (p. 29), we conclude that a significant amount of residue mass is missing in the Kim2018 coarse-grained structure as the domains that are identified by Kim et al. as "flexible linker beads" are not included in the scaffold structure. For example, in the case of Nup116, Nup100 and Nup145N, the 200-residue long non-cohesive domains are generally considered to be disordered [Yamada2010]. These linker domains are not included in the cryo-EM structure, nor are they in the simulated FG-Nup mesh. Hence, a significant percentage of the residue mass is still missing in the Kim2018 model simulation. The missing residues should be added to the Kim2018 structure (either as dynamic or rigid beads), such that the NPC mass is represented more accurately.

B. We note that Nup42 and Nup2 are missing in the Kim2018 model structure, while both FG-Nups contain an intrinsically disordered FG domain [Yamada2010, Kim2018]. We note that these two FG-Nups are in fact present in the NPC model of Huang et al. and expect that the authors clearly explain their choice for leaving out these two FG-Nups, and the implications this might have on their results. We also note that the Huang model contains an extra copy of Nup1. We ask the authors to critically compare their yeast NPC model with that of Huang et al.

C. The authors write "Thus, the disordered mesh region consisted of [...] amino acids, each being a continuous N-terminal fragment of the corresponding full-length protein." We note this is not always

correct, as the N-terminal domain of Nup159 (AA 1-381) has a well-defined beta-propeller structure [Denning2007, Yamada2010, Kim2018]. We suggest that the authors take a closer look at the relevant FG domains and also specify the exact FG-Nup segments (by stating the amino acid numbers and not only the length) that are used in the simulations.

D. The authors write "The C-terminal end of each FG-nup fragment was harmonically restrained to the anchor points prescribed by the Kim2018 structure." We stress to the authors that it is well-established that Nup1 and Nup60 are both anchored at their N-terminal end [Yamada2010, Mészáros2015, Kim2018].

****Response 1c)**

The authors have added new panels to Suppl. Fig. 2 to provide the requested charge-over-hydrophobicity ratios of all FG-Nup segments. The updated text addresses the observed differences in behavior between Nup145N in the Lin2016 model and Nup100+Nup116+Nup145N in the Kim2018 model.

****Response 1d)**

The authors' response to our doubts about the stated contact probabilities is not satisfactory. The authors have added the relative abundance of residues of each type in Suppl. Fig. 2b and f. We note that the suggested 44% abundance of hydrophobic residues in the yeast FG-Nups (Suppl. Fig. 2f) is in sharp contrast to the 20-25% hydrophobic residues in each FG-Nup as calculated by Yamada et al. The authors should clearly explain which amino acids they have considered as hydrophobic and why their values disagree with [Yamada2010]. Note that this might also have significant implications for the calculated C/H ratios.

Apart from this, we think that the results of Supplementary Fig. 2b and f are somewhat misleading. The authors show the probability that an inter-chain contact is between a specific pair of residue types (i.e. the percentage of measured contacts that are between that specific pair of residue types). However, this is not the same as "the probability of a pair of residue types to form an inter-chain contact" (from the caption), as here one needs to consider the relative abundance of each of the residue types.

Also, we point the authors to the fact that their analysis is incorrectly dealing with the permutations of the inter-chain contacts. In a multi-chain system, an inter-chain HP-contact is not the same as an inter-chain PH-contact. Hence, the fraction of HP contacts for a random mixture of residues (empty bars in Suppl. Fig 2b and f) should in fact be $F_{H,P} + F_{P,H}$ (i.e. doubled). Keeping this in mind, it is of course to be expected that there will be more HP (= HP + PH) contacts than HH or PP contacts.

We suggest that it would be much more informative to calculate the probability that a residue type is involved in inter-chain contacts (i.e. normalize for the abundance of hydrophobic and polar residues, respectively, and correctly handle possible permutations). One would then find that indeed the highest contact probability is for HH pairs (although the differences will be minor due to the large cut-off distance (0.8 nm) for defining an inter-residue contact).

****Response 2)**

The authors discuss the possible implications of the presence of a large amount of NTRs in the central mesh of the NPC and convincingly show that the effect on their results would be minor. The main text is correctly updated accordingly.

****Response 3)**

The authors provide sufficient evidence to support the claim that the apparent contrasting power laws

(of the MFPT with mass) might be caused by the limited ranges in mass considered in the different experiments.

**Related to our minor concerns on the initial document:

The authors have properly improved the manuscript to incorporate our minor concerns:

a) Supported by the added Supplementary Fig. 16, the authors have made clear what their reasoning is behind the estimated effective time step in their simulations. Based on the calculated gyration radii one could indeed presume the hydrodynamic interactions in the NPC FG-Nup to be negligible.

b) and c) We thank the authors for providing the extra simulation data that convincingly shows the correct implementation of the used coarse-grained model.

**Concerns based on the additions in the revised manuscript:

1) On page 7 the authors state that "Nup98 was observed to undergo phase separation at a density of about 250 mg/mL [Schmidt2015]." This is not correct: the concentration inside (phase separated) Nup98 particles is about 250 mg/mL; the critical concentration for Nup98 to phase separate is 1 to 50 µg/ml [Schmidt2015].

2) The authors state that they "did not observe any signs of phase separation" and base this on the calculated local densities being at most 130 mg/mL. However, the authors also state that in the FG-mesh about 65% of all FG-Nup chains are connected and the FG-Nups tethered to the NPC inner ring are almost all connected. One could argue that this high amount of inter-nup connections does suggest that some form of LLPS is taking place and that the lower local densities can be ascribed to the fact that the FG-Nups are anchored to the NPC scaffold. Can the authors elaborate more on why they think that there is no LLPS in their NPC models?

**There are several typos:

p.7 : "undergo a phase separation" -> "undergo phase separation", p.26 : "Nu49" -> "Nup49", p.28 : "Nsp" -> "Nsp1", p.28 : "copiesnof" -> "copies of".

**References used in this review:

[Denning2007] = Daniel P. Denning, Michael F. Rexach "Rapid evolution exposes the boundaries of domain structure and function in natively unfolded FG nucleoporins." *Molecular and Cellular Proteomics* 6 (2), 2010: 272–282.

[Huang2019] = Kai Huang, Mario Tagliazucchi, Sung Hyun Park, Yitzhak Rabin, Igal Szleifer "Nanocompartmentalization of the Nuclear Pore Lumen." *Biophysical Journal* 118, 2020: 219–231.

[Kim2018] = Seung Joong Kim, Javier Fernandez-Martinez, Ilona Nudelman, et al. "Integrative structure and functional anatomy of a nuclear pore complex." *Nature* 555 (7697), 2018: 475–482.

[Mészáros2015] = Noémi Mészáros, Jakub Cibulka, Maria Jose Mendiburo, Anete Romanuska, Maren Schneider, Alwin Köhler "Nuclear pore basket proteins are tethered to the nuclear envelope and can regulate membrane curvature." *Developmental Cell* 33 (3), 2015: 285–298.

[Schmidt2015] = Hermann Broder Schmidt, Dirk Görlich "Nup98 FG domains from diverse species spontaneously phase-separate into particles with nuclear pore-like permselectivity." *eLife* 4, 2015: e04251.

[Yamada2010] = Justin Yamada, Joshua L. Phillips, Samir Patel, et al. "A bimodal distribution of two distinct categories of intrinsically disordered structures with separate functions in FG nucleoporins." *Molecular and Cellular Proteomics* 9 (10), 2010: 2205–2224.

Reviewer #3:

Remarks to the Author:

I have read the authors' (long) rebuttal document. It seems to me that they have done a commendable job of responding to all concerns, in many cases by doing additional computations on modified structural or kinetic models.

The system under study is very large and very complicated, with many atomistic level details being unresolved at present. Given these conditions, one cannot expect completely rigorous conclusions. Nevertheless, having carefully considered the many suggestions for system modification provided by the reviewers, the essential conclusions of the paper have not changed relative to the initial submission. Overall, the authors have advanced the state of our understanding of passive diffusion of globular proteins through the NPC significantly. I recommend publication of this paper in its present form, subject to (presumably) minor modifications that the other referees may suggest. I have no further requests of this type.

RESPONSE TO REVIEWER COMMENTS

Reviewer #1 (Remarks to the Author):

The authors have answered all my requests. In particular the comparison with the state of the art on nuclear pore modelling have been strengthened which highlights the novelty of this work for experimentalists. I think this new version of the manuscript will be a valuable tool for the understanding of nucleo-cytoplasmic transport.

For these reasons I strongly support the publication of this work in Nature Communications.

We thank Reviewer 1 for the enthusiastic evaluation of our work.

Reviewer #2 (Remarks to the Author):

Below is our review of the revised manuscript of NCOMMS-21-49443A: Percolation transition determines protein size limit for passive transport through the nuclear pore complex

The authors have added new results on a yeast NPC structure (Kim et al) to answer our main concerns on the low FG-Nup mass in the initial composite structure (Lin et al). However, there are several issues with the assumptions made in this new computational model related to mass inclusion and the performed contact analysis. The authors should properly remedy these before the paper can be accepted for publication.

The review below is numbered according to the answers the authors gave (named 'Response 1a, 1b, etc.') to the initial review (reviewer #2) in their rebuttal letter.

****Response 1a)**

We thank the authors for taking over our suggestion to describe the Lin2016 model as 'composite' rather than 'complete'.

We are happy to find that our revisions have satisfied this specific concern.

****Response 1b)**

The authors have tried to address our concerns about their reported NPC densities by repeating their analyses on a second, more complete NPC model (the Kim2018 model).

However, we have several remarks on the added section that discusses the Kim2018 coarse-grained structure:

We thank Reviewer 2 for the careful evaluation of our revised manuscript and for additional remarks that we have addressed in the revised manuscript as described below.

A. Based on the supplied FG-Nup sequence lengths (p. 29), we conclude that a significant amount of residue mass is missing in the Kim2018 coarse-grained structure as the domains that are identified by Kim et al. as “flexible linker beads” are not included in the scaffold structure. For example, in the case of Nup116, Nup100 and Nup145N, the 200-residue long non-cohesive domains are generally considered to be disordered [Yamada2010]. These linker domains are not included in the cryo-EM structure, nor are they in the simulated FG-Nup mesh. Hence, a significant percentage of the residue mass is still missing in the Kim2018 model simulation. The missing residues should be added to the Kim2018 structure (either as dynamic or rigid beads), such that the NPC mass is represented more accurately.

In response to this comment, we have created an updated model of a yeast NPC that we refer to as “Kim2018+”. Non-cohesive connector domains have been added for Nup116 (AA 761-965), Nup100 (AA 561-815) and Nup145N (AA 221-458). Sixteen of each linker ‘connector’ domains were added, following the stoichiometry of our original model. In our simulations, these domains were restrained to their N- and C-termini and were observed to reside primarily within the NPC scaffold.

B. We note that Nup42 and Nup2 are missing in the Kim2018 model structure, while both FG-Nups contain an intrinsically disordered FG domain [Yamada2010, Kim2018]. We note that these two FG-Nups are in fact present in the NPC model of Huang et al. and expect that the authors clearly explain their choice for leaving out these two FG-Nups, and the implications this might have on their results. We also note that the Huang model contains an extra copy of Nup1. We ask the authors to critically compare their yeast NPC model with that of Huang et al.

Thank you for pointing out this omission. In the updated Kim2018+ model, we have elected to include eight copies of Nup42 (AA 1-382) and sixteen copies of Nup2 (AA 1-720). We set the length, stoichiometry, anchor radial position and anchor height along z to match the NPC model of Huang et al.

To facilitate systematic comparison of our Kim2018 and Kim2018+ models of yeast NPC to that of Huang2020, we added the following to the SI as Supplementary Table 3:

FG-nup species	Kim2018 copy #	Kim2018 length	Kim2018 domain	Kim2018+ copy #	Kim2018+ length	Kim2018+ domain	Huang2020 copy #	Huang2020 length	Huang2020 domain
Nsp1	48	620	1-620	48	620	1-620	48	601	1-601
Nup159	16	760	1-760	16	760	382-1141	16	695	388-1082
Nup116	16	760	1-760	16	760	1-760	16	966	1-966
Nup100	16	560	1-560	16	560	1-560	16	801	1-801
Nup49	32	220	1-220	32	220	1-220	32	270	1-270
Nup57	32	220	1-220	32	220	1-220	32	287	1-287
Nup145N	16	220	1-220	16	220	1-220	16	426	1-426
Nup1	8	720	1-720	8	720	1076-357	16	876	1076-201
Nup60	16	120	1-120	16	120	539-420	16	189	539-351
Nup42	-	-	-	8	382	1-382	8	382	1-382
Nup2	-	-	-	16	720	1-720	16	720	1-720
linker Nup100	-	-	-	16	225	561-815	-	-	-
linker Nup116	-	-	-	16	205	761-965	-	-	-
linker Nup145N	-	-	-	16	238	221-458	-	-	-

The following text was added to the methods section “CG models of a yeast NPC”:
 “We note that our Kim2018 yeast NPC model does not include FG-nups Nup42 or Nup2, which are present in the Huang2020 model. Furthermore, FG-nups Nup1 and Nup60 are anchored from the C-terminal end in Kim2018 but from their N-terminal end in Huang2020. Notably, Huang2020 contains sixteen copies of Nup1, whereas our Kim2018 model only has eight. The starting point of Nup159 is different for Kim2018 (residue 1) versus Huang2020 (residue 388). Lastly, the domains of Nup100/Nup116/Nup145N are quite a bit shorter for Kim2018 compared to Huang2020. Combined, these differences mean the total mass of the FG-nup domains of the Huang2020 yeast NPC model is about 30% greater than the total mass of the domains present in our original Kim2018 model.

“The updated yeast NPC model, which we refer to as Kim2018+, is much more closely related to Huang2020 than our original Kim2018 model. FG-nups Nup42 and Nup2 are included in our Kim2018+ model, with the same stoichiometry and domain lengths as in Huang2020. Furthermore, linker domains of Nup100/Nup116/Nup145N are present in our Kim2018+ model, increasing those nups’ lengths to similar lengths as present in Huang2020. Nup1 and Nup60 are tethered from their N-terminal ends for Kim2018+, as is done in Huang2020. And the starting points of Nup159 are close for Kim2018+ (residue 382) and Huang2020 (residue 388). Overall, the total mass of the FG-nup domains present in our updated Kim2018+ model is quite close to the total mass of Huang2020.”

C. The authors write “Thus, the disordered mesh region consisted of [...] amino acids, each being a continuous N-terminal fragment of the corresponding full-length protein.” We note this is not always correct, as the N-terminal domain of Nup159 (AA 1-381) has

a well-defined beta-propeller structure [Denning2007, Yamada2010, Kim2018]. We suggest that the authors take a closer look at the relevant FG domains and also specify the exact FG-Nup segments (by stating the amino acid numbers and not only the length) that are used in the simulations.

We thank the reviewer for their attention to detail. For the Kim2018+ NPC model, the domain was updated to Nup159 (AA 382-1141). Methods sections for our Lin2016, Kim2018 and Kim2018+ models have been updated to explicitly state amino acid domains included for each nup species.

D. The authors write “The C-terminal end of each FG-nup fragment was harmonically restrained to the anchor points prescribed by the Kim2018 structure.” We stress to the authors that it is well-established that Nup1 and Nup60 are both anchored at their N-terminal end [Yamada2010, Mészáros2015, Kim2018].

For the Kim2018+ NPC model, the following domains were included for Nup1 (AA 1076-357) and Nup60 (AA 539-420). And they were anchored at their N-termini.

In response to points A–D, we built and simulated another coarse-grained yeast NPC starting from Kim et al.’s scaffold to more closely match what is present in the yeast NPC model simulated by Huang et al.

We highlight the differences between our two coarse-grained yeast NPC models, and their FG-nup densities in Supplementary Figure 15, which we reproduce below:

We describe this Kim2018+ yeast NPC model in the updated methods as follows:

“Another version of a yeast NPC was designed separately, which we refer to as “Kim2018+”. We employed the same nuclear envelope and scaffold potentials as was done for our Kim2018 model. The nup species, and their domains, for Kim2018+ were as follows: Nsp1 (AA 1– 620), Nup159 (AA 382–1141), Nup116 (AA 1–760), Nup100 (AA 1–560), Nup49 (AA 1–220), Nup57 (AA 1–220), Nup145N (AA 1–220), all anchored at their N-terminal ends; Nup1 (AA 1076–357) and Nup60 (AA 539–420), which were anchored at their C-terminal ends; previously unconsidered nup species Nup42 (eight copies, AA 1–382), Nup2 (sixteen copies, AA 1–720); and linker ‘connector’ domains Nup116 (AA 561–815), Nup100 (AA 761–965), Nup145N (AA 221–458), whose N- and C-terminal ends were restrained but were otherwise fully flexible. Supplementary Table 3 compares the FG-nup species’ copy-numbers, lengths and amino-acid ranges present for NPC models in our paper (Kim2018, Kim2018+) and one from Huan et al., which we refer to as “Huan2020.””

And the following paragraph was added to the Results of the main text:

“We built another coarse-grained representation of the yeast NPC based on the integrative scaffold constructed by Kim et al., inspired, in part, by the composition of a yeast

NPC model simulated by Huang et al., which we refer to as “Huang2020.” As shown in Supplementary Fig. 15a–c, compared to our Kim2018 model, two new FG-nup species were included—Nup42, Nup2—and flexible linker regions restrained only on their terminal ends for Nup116, Nup100, Nup145N in the version we call “Kim2018+.” In all, the FG-nup domains of Kim2018+ have about 30% more amino acid residues present than our Kim2018, and the overall FG-nup mass of Kim2018+ is quite close to that of Huang2020. The exact compositions of these models (Kim2018, Kim2018+, Huang2020) are provided in Supplementary Table 3. From Supplementary Fig. 15d–f, we observe that the additional FG-nup domains in Kim2018+ lead to a slightly increased density within the central channel and on the nucleoplasmic side (i.e. negative in z). The difference in density, however, does not exceed 30 mg/mL, and it is less than 5 mg/mL in several regions of the NPC volume, Supplementary Fig. 15f. We anticipate, therefore, that the PMF barrier heights would be slightly higher, and MFPT slightly longer, for Kim2018+ compared to Kim2018. Furthermore, we expect the percolation transition that corresponds to a power law-to-exponential change in MFPT scaling behavior would occur at a slightly smaller protein size for Kim2018+ compared to Kim2018. We expect the differences would be relatively small between these two models, however.”

**Response 1c)

The authors have added new panels to Suppl. Fig. 2 to provide the requested charge-over-hydrophobicity ratios of all FG-Nup segments. The updated text addresses the observed differences in behavior between Nup145N in the Lin2016 model and Nup100+Nup116+Nup145N in the Kim2018 model.

We are happy to find that our revisions have satisfied this specific concern.

**Response 1d)

The authors’ response to our doubts about the stated contact probabilities is not satisfactory. The authors have added the relative abundance of residues of each type in Suppl. Fig. 2b and f. We note that the suggested 44% abundance of hydrophobic residues in the yeast FG-Nups (Suppl. Fig. 2f) is in sharp contrast to the 20-25% hydrophobic residues in each FG-Nup as calculated by Yamada et al. The authors should clearly explain which amino acids they have considered as hydrophobic and why their values disagree with [Yamada2010]. Note that this might also have significant implications for the calculated C/H ratios.

This difference arose over the definition of “hydrophobic residues.” We originally chose hydrophobic to mean ‘AILFWVMYGP’, whereas [Yamada2010] uses ‘AILFWV’. The

additional residue types 'MYGP' lead to the greater abundance. Following the definition of "hydrophobic" stated in [Yamada2010], we have recalculated the percentage of hydrophobic residues in Lin2016 (24%) and Kim2018 (24%), within the range of 20-25%. C/H ratios have also been updated using 'AILFWV' to define hydrophobic.

Apart from this, we think that the results of Supplementary Fig. 2b and f are somewhat misleading. The authors show the probability that an inter-chain contact is between a specific pair of residue types (i.e. the percentage of measured contacts that are between that specific pair of residue types). However, this is not the same as "the probability of a pair of residue types to form an inter-chain contact" (from the caption), as here one needs to consider the relative abundance of each of the residue types.

Also, we point the authors to the fact that their analysis is incorrectly dealing with the permutations of the inter-chain contacts. In a multi-chain system, an inter-chain HP-contact is not the same as an inter-chain PH-contact. Hence, the fraction of HP contacts for a random mixture of residues (empty bars in Suppl. Fig 2b and f) should in fact be $F_{H,P} + F_{P,H}$ (i.e. doubled). Keeping this in mind, it is of course to be expected that there will be more HP (= HP + PH) contacts than HH or PP contacts.

We suggest that it would be much more informative to calculate the probability that a residue type is involved in inter-chain contacts (i.e. normalize for the abundance of hydrophobic and polar residues, respectively, and correctly handle possible permutations). One would then find that indeed the highest contact probability is for HH pairs (although the differences will be minor due to the large cut-off distance (0.8 nm) for defining an inter-residue contact).

Thank you for the suggestions. To facilitate comparison to the previous work, we have defined the following categories of amino-acid residue types: Hydrophobic 'AILFWV', Polar 'STNQH', Charged 'DEKR', Other 'MYGPC', abbreviated H, P, C, O. We have chosen to show a single bar for each type, meaning contacts with that type involved. And we have estimated the percentage of all contacts by considering all permutations. For hydrophobic "H" for example: HH, HP, PH, HC, CH, HO, OH.

Below we provide the updated Supplementary Figure on Properties of the FG-nup mesh:

For the sake of comparison, below we also include the previous version of the same Supplementary figure:

We note panels a & e have remained the same. Our approach to contacts has changed from pairwise to type involved in a contact. C/H ratios are slightly higher in our new plot, because the updated definition of hydrophobic "H" involves fewer residue types than before. Hydrophobic maps are lower in overall density with the updated definition, but qualitatively similar to how they appeared previously.

****Response 2)**

The authors discuss the possible implications of the presence of a large amount of NTRs in the central mesh of the NPC and convincingly show that the effect on their results would be minor. The main text is correctly updated accordingly.

We are happy to find that our revisions have satisfied this specific concern.

****Response 3)**

The authors provide sufficient evidence to support the claim that the apparent contrasting power laws (of the MFPT with mass) might be caused by the limited ranges in mass considered in the different experiments.

We are happy to find that our revisions have satisfied this specific concern.

****Related to our minor concerns on the initial document:**

The authors have properly improved the manuscript to incorporate our minor concerns:

a) Supported by the added Supplementary Fig. 16, the authors have made clear what their reasoning is behind the estimated effective time step in their simulations. Based on the calculated gyration radii one could indeed presume the hydrodynamic interactions in the NPC FG-Nup to be negligible.

b) and c) We thank the authors for providing the extra simulation data that convincingly shows the correct implementation of the used coarse-grained model.

We are happy to find that our revisions have satisfied these three specific concerns.

****Concerns based on the additions in the revised manuscript:**

1) On page 7 the authors state that “Nup98 was observed to undergo phase separation at a density of about 250 mg/mL [Schmidt2015].” This is not correct: the concentration inside (phase separated) Nup98 particles is about 250 mg/mL; the critical concentration for Nup98 to phase separate is 1 to 50 µg/ml [Schmidt2015].

We apologize for that incorrect statement. We have chosen to remove it to avoid any confusion.

2) The authors state that they “did not observe any signs of phase separation” and base this on the calculated local densities being at most 130 mg/mL. However, the authors also state that in the FG-mesh about 65% of all FG-Nup chains are connected and the FG-Nups tethered to the NPC inner ring are almost all connected. One could argue that this high amount of inter-nup connections does suggest that some form of LLPS is taking place and that the lower local densities can be ascribed to the fact that the FG-Nups are anchored to the NPC scaffold. Can the authors elaborate more on why they think that there is no LLPS in their NPC models?

We agree with the reviewer that the FG-nup mesh is highly connected together in what can be considered a kind of phase separation. Because the environment is constrained by the NPC scaffold, and the phase occurs on the nanoscale, we hesitate to refer to this as an example of liquid–liquid phase separation and would rather call it “nanophase separation.”

In response, we have added the following sentences to the main text:

“Within the context of the NPC, FG-nups condense together at the nanoscale, which we consider to be related, but distinct, from the liquid–liquid phase separation that drives the formation of membraneless organelles. Constrained by the NPC scaffold with interactions programmed by their specific sequences, FG-nup domains are highly connected together and undergo what Huang et al. refer to as “nanophase separation.”

**There are several typos:

p.7 : “undergo a phase separation” -> “undergo phase separation”, p.26 : “Nu49” -> “Nup49”, p.28 : “Nsp” -> “Nsp1”, p.28 : “copiesnof” -> “copies of”.

Thank you for pointing out these typos. We have corrected them.

**References used in this review:

[Denning2007] = Daniel P. Denning, Michael F. Rexach “Rapid evolution exposes the boundaries of domain structure and function in natively unfolded FG nucleoporins.”

Molecular and Cellular Proteomics 6 (2), 2010: 272–282.

[Huang2019] = Kai Huang, Mario Tagliacruzchi, Sung Hyun Park, Yitzhak Rabin, Igal Szleifer “Nanocompartmentalization of the Nuclear Pore Lumen.” Biophysical Journal 118, 2020: 219–231.

[Kim2018] = Seung Joong Kim, Javier Fernandez-Martinez, Ilona Nudelman, et al. “Integrative structure and functional anatomy of a nuclear pore complex.” Nature 555 (7697), 2018: 475–482.

[Mészáros2015] = Noémi Mészáros, Jakub Cibulka, Maria Jose Mendiburo, Anete Romanauska, Maren Schneider, Alwin Köhler “Nuclear pore basket proteins are tethered to the nuclear envelope and can regulate membrane curvature.” *Developmental Cell* 33 (3), 2015: 285–298.

[Schmidt2015] = Hermann Broder Schmidt, Dirk Görlich “Nup98 FG domains from diverse species spontaneously phase-separate into particles with nuclear pore-like permselectivity.” *eLife* 4, 2015: e04251.

[Yamada2010] = Justin Yamada, Joshua L. Phillips, Samir Patel, et al. “A bimodal distribution of two distinct categories of intrinsically disordered structures with separate functions in FG nucleoporins.” *Molecular and Cellular Proteomics* 9 (10), 2010: 2205–2224.

Reviewer #3 (Remarks to the Author):

I have read the authors' (long) rebuttal document. It seems to me that they have done a commendable job of responding to all concerns, in many cases by doing additional computations on modified structural or kinetic models.

The system under study is very large and very complicated, with many atomistic level details being unresolved at present. Given these conditions, one cannot expect completely rigorous conclusions. Nevertheless, having carefully considered the many suggestions for system modification provided by the reviewers, the essential conclusions of the paper have not changed relative to the initial submission. Overall, the authors have advanced the state of our understanding of passive diffusion of globular proteins through the NPC significantly. I recommend publication of this paper in its present form, subject to (presumably) minor modifications that the other referees may suggest. I have no further requests of this type.

We thank Reviewer 3 for the careful evaluation of our revised manuscript and for the favorable recommendation.